# Neocortical pyramidal neurons with axons emerging from dendrites are frequent in non-primates, but rare in monkey and human

**Petra Wahle[1]\*, Eric Sobierajski[1†], Ina Gasterstädt[1†], Nadja Lehmann[2], Susanna Weber[2], Joachim HR Lübke[3], Maren Engelhardt[4], Claudia Distler[5], Gundela Meyer[6]**

[1]Ruhr University Bochum, Faculty of Biology and Biotechnology, Developmental Neurobiology, Bochum, Germany; [2]Heidelberg University, Medical Faculty Mannheim, Mannheim Center for Translational Neuroscience, Institute of Neuroanatomy, Mannheim, Germany; [3]JARA-Institute Brain Structure Function Relationship, Jülich, Germany; [4]Johannes Kepler University Linz, Faculty of Medicine, Institute of Anatomy and Cell Biology, Linz, Austria; [5]Ruhr University Bochum, Faculty of Biology and Biotechnology, Zoology and Neurobiology, Bochum, Germany; [6]University of La Laguna, Faculty of Medicine, Department of Basic Medical Science, Santa Cruz de Tenerife, Spain

**\*For correspondence:**
petra.wahle@rub.de

[†]These authors contributed equally to this work

**Competing interest:** The authors declare that no competing interests exist.

**Abstract** The canonical view of neuronal function is that inputs are received by dendrites and somata, become integrated in the somatodendritic compartment and upon reaching a sufficient threshold, generate axonal output with axons emerging from the cell body. The latter is not necessarily the case. Instead, axons may originate from dendrites. The terms 'axon carrying dendrite' (AcD) and 'AcD neurons' have been coined to describe this feature. In rodent hippocampus, AcD cells are shown to be functionally 'privileged', since inputs here can circumvent somatic integration and lead to immediate action potential initiation in the axon. Here, we report on the diversity of axon origins in neocortical pyramidal cells of rodent, ungulate, carnivore, and primate. Detection methods were Thy-1-EGFP labeling in mouse, retrograde biocytin tracing in rat, cat, ferret, and macaque, SMI-32/βIV-spectrin immunofluorescence in pig, cat, and macaque, and Golgi staining in macaque and human. We found that in non-primate mammals, 10–21% of pyramidal cells of layers II–VI had an AcD. In marked contrast, in macaque and human, this proportion was lower and was particularly low for supragranular neurons. A comparison of six cortical areas (being sensory, association, and limbic in nature) in three macaques yielded percentages of AcD cells which varied by a factor of 2 between the areas and between the individuals. Unexpectedly, pyramidal cells in the white matter of postnatal cat and aged human cortex exhibit AcDs to much higher percentages. In addition, interneurons assessed in developing cat and adult human cortex had AcDs at type-specific proportions and for some types at much higher percentages than pyramidal cells. Our findings expand the current knowledge regarding the distribution and proportion of AcD cells in neocortex of non-primate taxa, which strikingly differ from primates where these cells are mainly found in deeper layers and white matter.

## Editor's evaluation

Wahle and colleagues show that excitatory cortical neurons differ in the fundamental structural arrangement of dendrites, soma and axons across a range of mammalian species. Axons can originate directly from pyramidal cell dendrites in species as diverse as rodents, ferret, cats, pigs and primates. However, cross-species comparisons also indicate that non-primate brains have a higher proportion of axon-carrying-dendrites (AcD) than did brains of primates. This paper is of potential interest to a broad range of neuroscientists working on brain structure and function as well as computational models thereof.

## Introduction

The prevailing concept of neocortical pyramidal cell function proposes that excitatory inputs arrive via the dendrites, are integrated in the somatodendritic compartment, and upon reaching sufficient threshold, the axonal domain generates an action potential. The axon usually originates from the ventral aspect of the soma, starting with a short axon hillock followed by the axon initial segment (AIS), the electrogenic domain generating the action potential (reviewed by *Kole and Brette, 2018*). Already Ramon y Cajal suggested that impulses may bypass the soma and flow directly to the axon (reviewed by *Triarhou, 2014*). Axon carrying dendrites (AcDs) are common in cortical inhibitory interneurons (*Meyer, 1987*; *Wahle and Meyer, 1987*; *Meyer and Wahle, 1988*; *Höfflin et al., 2017*). Furthermore, upright, inverted, and fusiform pyramidal neurons of supra- and infragranular layers display AcDs in Golgi impregnated or dye-injected cortex from rodents, lagomorphs, ungulates, and carnivores (*Peters et al., 1968*; *Smit and Uylings, 1975*; *van der Loos, 1976*; *Peters and Kara, 1985*; *Ferrer et al., 1986a*; *Ferrer et al., 1986b*; *Hübener et al., 1990*; *Reblet et al., 1992*; *Matsubara et al., 1996*; *Prieto and Winer, 1999*; *Mendizabal-Zubiaga et al., 2007*; *Hamada et al., 2016*; *Ernst et al., 2018*). In mouse hippocampal CA1 pyramidal cells, axons frequently emerge from basal dendrites (*Thome et al., 2014*). Multiphoton glutamate uncaging and patch clamp recordings revealed that input to the AcD is more efficient in eliciting an action potential than input onto regular dendrites (non-AcDs). AcDs are intrinsically more excitable, generating dendritic spikes with higher probability and greater strength. Synaptic input onto AcDs generates action potentials with lower thresholds compared to non-AcDs, presumably due to the short electrotonic distance between input and the AIS. The anatomical diversity of axon origins plus the diversity of length and position of the AIS substantially impact the electrical behavior of pyramidal neurons (reviewed by *Kole and Brette, 2018*). This begs the questions, of how frequent AcD pyramidal neurons are among the mammalian species, and whether AcD pyramidal neurons also exist in primates. Our data suggest remarkable differences between phylogenetic orders and position in gray and white matter.

## Results

### Pyramidal AcD cells in adult cortex

We assigned AcDs in a very conservative manner. All cells in which the axonal origin could not be unequivocally seen to arise from a dendrite were considered 'somatic' axon cells. A certain fraction of neurons had an axon which shares a root with a dendrite. We consequently considered 'shared root' cells as somatic axon cells. *Figure 1A* documents the diversity of axon origins of pyramidal cells in a P60 infant macaque monkey cortex with an axon originating from a soma (inset B), an AcD (inset C), and the shared root configuration (inset D). Generally, AcDs were basal dendrites. AcD neurons of other species are shown in *Figure 2A–E*. Macaque neurons stained with SMI-32/βIV-spectrin are shown in three videos. *Figure 2—video 1* shows an AcD pyramidal neuron from premotor cortex and flanking non-AcD neurons. *Figure 2—video 2* shows a layer V pyramidal cell of the cingulate cortex with an axon wiggling out between thick dendrites at the right somatic pole. Even with the help of confocal imaging it was not easy to unequivocally identify the origin of the axon, and we score these neurons as a shared root cell. *Figure 2—video 3* shows a spindle-shaped neuron resembling a Von Economo neuron of infragranular layers of the cingulate cortex. Note that the axon emerges >65 μm away from the soma from the descending dendrite.

The online version of this article includes the following figure supplement for *Figure 2*:

To further demonstrate the variability of the axon origin, *Figure 2—figure supplement 1A-L* shows representative biocytin-labeled neurons of rat, ferret, and macaque. The shared root configuration can

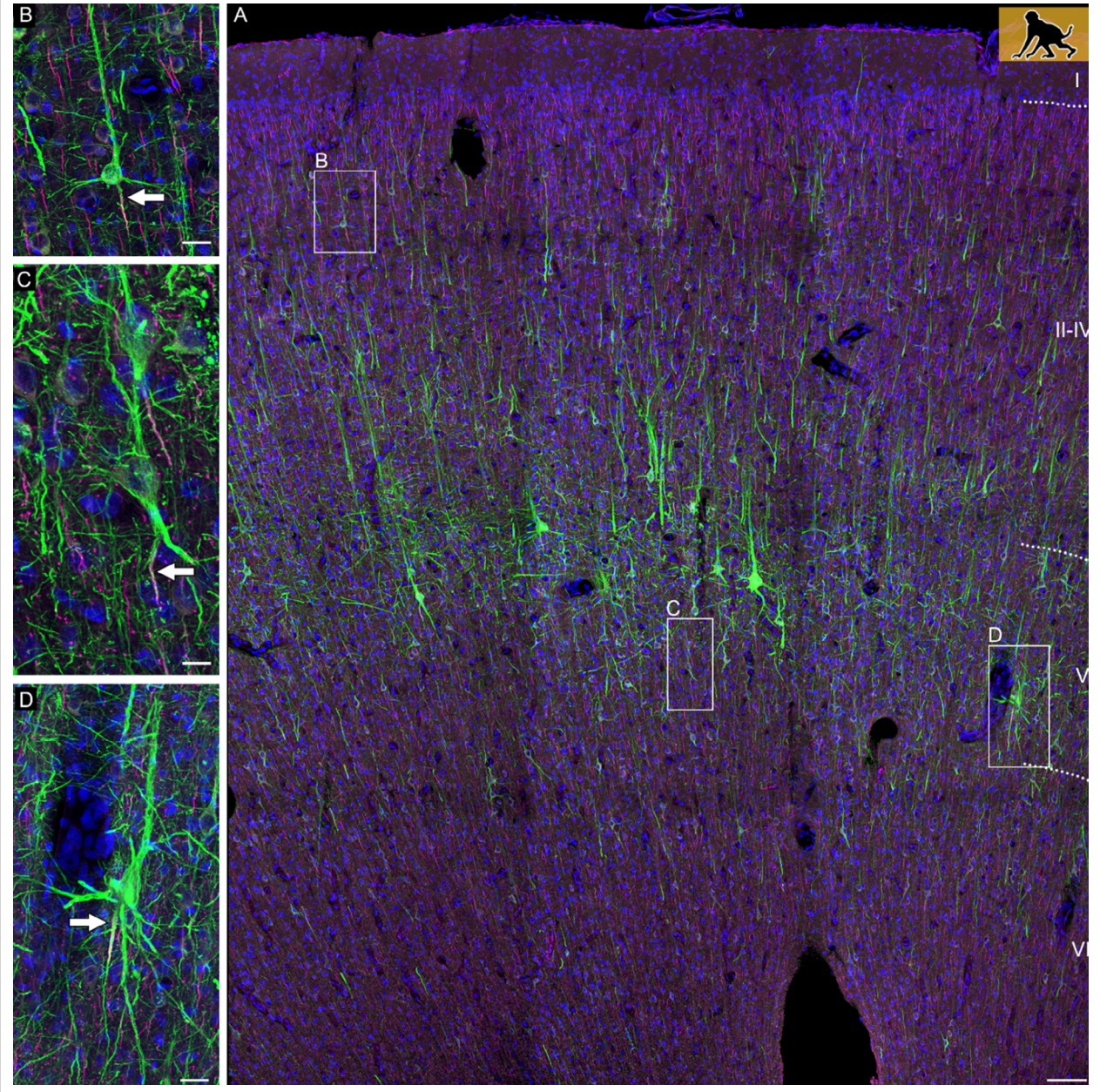

**Figure 1.** Confocal tile scan of dorsal neocortex (premotor area) of P60 infant macaque. (**A**) Pyramidal cells were stained with SMI-32/βIV-spectrin to label dendrites and the axon initial segment, respectively. Insets depict neurons with an axon emerging from the soma (**B**), or from an axon carrying dendrite (**C**), or a shared root (**D**). Axons indicated by arrows. Scale bars 100 µm for the tile scan and 25 µm for the insets.

be considered a transition between a clear-cut somatic axon and an axon originating from a dendrite. Neurons with the three features co-occur: for instance, of the group of rat pyramidal neurons depicted in *Figure 2—figure supplement 1A* and at higher magnifications in *Figure 2—figure supplement 1B-D*, some have somatic axons (cells labeled 1, 3), one has a shared root (cell 2) and two have AcDs (cells 4, 5). *Figure 2—figure supplement 2A-O* shows Golgi-impregnated pyramidal neurons of human cortex. The black reaction product made it more difficult to identify unequivocal AcD cells.

## Quantitative analysis

In gray matter of non-primates, 10–21% of the pyramidal neurons assessed by perpendicular counts through all layers had an AcD (*Figure 3A*). The interindividual variability and staining methods are reported for mouse and rat in *Table 1*, and for cat, ferret, and pig in *Table 2*. Mouse Thy-1-EGFP,

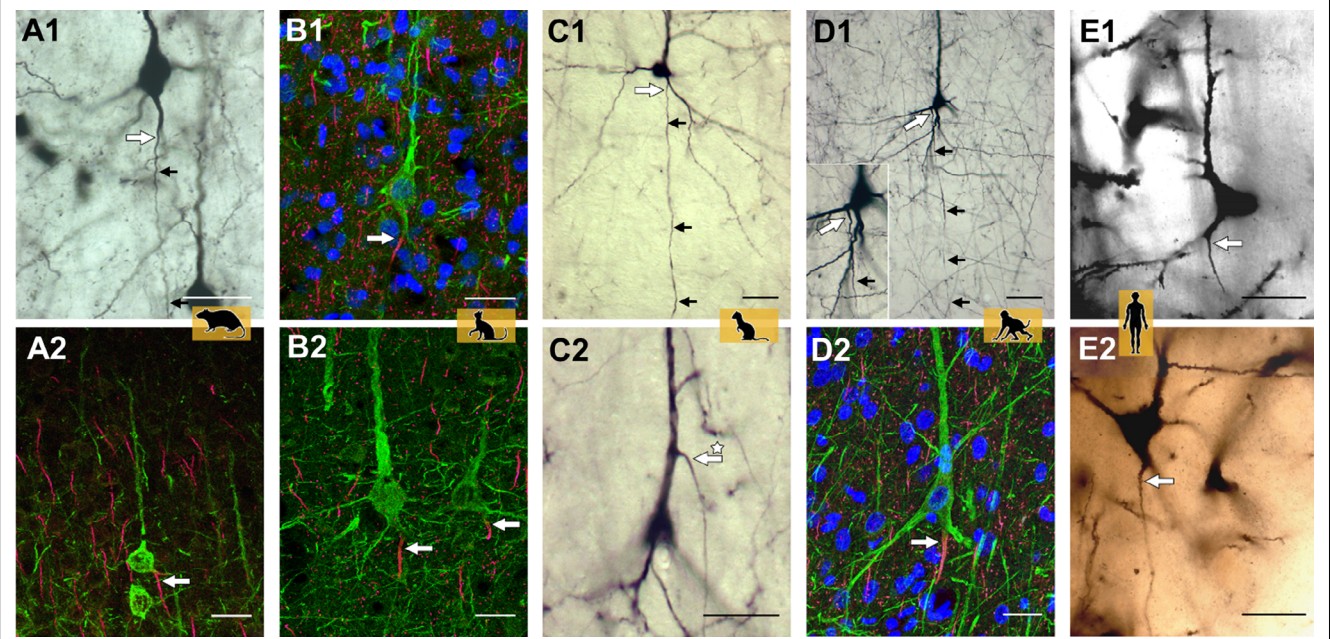

**Figure 2.** Representative axon carrying dendrite (AcD) neurons. (**A1, A2**) From rat visual cortex (biocytin, immunofluorescence); (**B1, B2**) cat visual cortex (immunofluorescence); (**C1, C2**) ferret visual cortex (biocytin); (**D1, D2**) macaque premotor cortex (biocytin, immunofluorescence), the inset shows the axon origin at higher magnification; (**E1, E2**) human auditory cortex (Golgi method; D2 is a montage of two photos). Apical AcDs (asterisk in C2) were rare, less than 10 were detected among the neurons assessed in adult rat, ferret, and macaque, and none in our human material. In all cases, the axon immediately bent down toward the white matter. Axon origins are marked by large arrows, small arrows indicate the course of biocytin-labeled axons. Scale bars 25 μm.

The online version of this article includes the following video and figure supplement(s) for figure 2:

**Figure supplement 1.** Variations of axon origins of biocytin-stained pyramidal neurons of rat (**A–D**) and ferret (**E–G**) visual cortex, and macaque premotor cortex (**H–K**), and macaque intraparietal sulcus (**L**).

**Figure supplement 2.** Variations of axon origins of Golgi-impregnated pyramidal neurons of human temporal cortex, all taken from Individual 2, 56 years of age and sampled from all layers.

**Figure 2—video 1.** Neuron classified as AcD cell in macaque cortex.
https://elifesciences.org/articles/76101/figures#fig2video1

**Figure 2—video 2.** Neuron classified as shared root cell in macaque cortex.
https://elifesciences.org/articles/76101/figures#fig2video2

**Figure 2—video 3.** Neuron classified as Von Economo neuron in macaque cortex.
https://elifesciences.org/articles/76101/figures#fig2video3

βIV-spectrin-positive pyramidal neurons vary from 10 to 22%, possibly due to the individual variability of the Thy-1 expression level. In adult macaque, gray matter of only about 3–6% of the pyramidal neurons had an AcD (*Figure 3A*). The interindividual variability and staining methods are reported in *Table 3*. In human gray matter, the proportion of AcD pyramidal neurons was 1.96% on average. The interindividual variability and staining methods are reported in *Table 4*.

A significant difference emerged after a layer-specific analysis. Proportions were largely obtained in a second round of quantification with surface-parallel tracks, and in some cases sections were assessed that had not been analyzed in the first round of counting in order to obtain higher cell numbers. Therefore, in *Tables 1–4* the laminar percentages do not simply add up to proportions obtained for whole gray matter. Furthermore, we plotted the individual values because bar graphs do not represent interindividual variability. Non-primates had about equal proportions of AcD cells in supra- and infragranular layers (*Figure 3B*, *Tables 1 and 2*). Thy-1 was only expressed in layer V and therefore, did not allow to determine laminar percentages for mouse. The macaque had only about 1–5% supragranular and about 5–14% infragranular AcD cells (*Figure 3B*, *Table 3*). Note the variable proportions of AcD neurons in infragranular layers and no obvious correlation between proportion

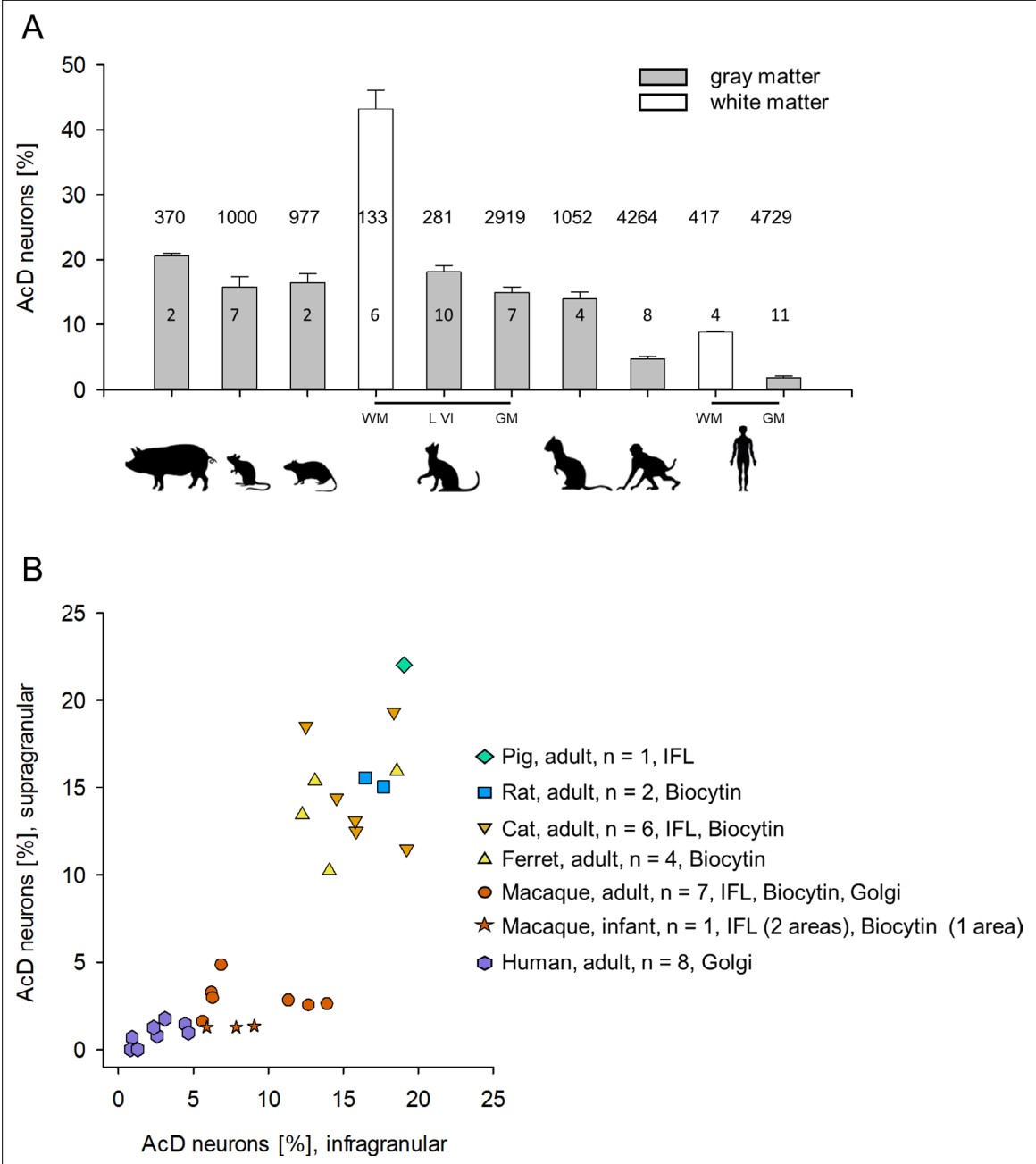

**Figure 3.** Proportion of axon carrying dendrite (AcD) neurons across species. (**A**) Shown are mean ± SEM. of the individual percentages listed in *Tables 1–4*, which also indicate the staining methods. Numbers above the bars are the total number of pyramidal neurons assessed per species/ cell class for this graph. Numbers in the bars indicate the number of individuals. (**B**) Laminar analysis. Non-primate species showed roughly equal proportions of AcD neurons in supra- and infragranular layers. With some individual variability the range was 10–21%. In contrast, in macaque, the cluster was downshifted along the ordinate due to overall much lower proportions. Furthermore, infragranular pyramidal cells displayed much higher proportions of AcD cells compared with supragranular pyramidal cells. A Mann-Whitney rank sum test of 'all non-primate' versus 'all macaque' percentages of supragranular and infragranular AcD cells, yielded p<0.001 and p<0.001, respectively. Human was not included in the statistical test because only one method was used to detect AcD cells. The legend indicates the number of individuals and the staining methods; IFL, immunofluorescence. Note that we could not do a laminar analysis for all individuals shown in (**A**) because staining of supragranular layers in some animals delivered too low numbers which might have led to a sampling error.

The online version of this article includes the following source data for figure 3:

**Source data 1.** Data and statistical analysis of experiments shown in *Figure 3A, B*.

**Table 1.** Proportion of pyramidal neurons with AcD: rodents.

| Species; cortical area; staining method; age; sex | Proportion of AcD cells [%]; n of cells assessed | By layers: supra %, infra % |
|---|---|---|
| **Mouse S1 cortex, layer V;** **Thy1-EGFP/βIV-spectrin immunofluorescence** | | |
| Adult, female | 17.42%, 178 cells | n.a. |
| Adult, female | 11.78%, 348 cells | n.a. |
| Adult, female | 15.79%, 36 cells | n.a. |
| Adult, female | 18.84%, 138 cells | n.a. |
| Adult, male | 21.93%, 187 cells | n.a. |
| Adult, male | 14.82%, 54 cells | n.a. |
| Adult, male | 10.17%, 59 cells | n.a. |
| *Average [%], total n of cells* | **15.82%, 1000** cells | |
| **Rat visual cortex; biocytin tracing** | | |
| Adult, male, two hemispheres | 17.82%, 174 cells | 15.06%, 17.69% |
| Adult, male, two hemispheres | 15.07%, 803 cells | 15.65%, 16.46% |
| *Average [%], total n of cells* | **16.45%, 977** cells | **15.36%, 17.08%** |

AcD, axon carrying dendrite; n.a., not applicable.

and age of the individual macaques. The differences between non-primates and macaques were significant (see legend to *Figure 3B*). Values obtained in human Golgi material overlap with the lower range of the macaque values. Also in humans, supragranular layers had low proportions of AcD cells of 0.99% on average. Laminar percentages for infragranular neurons were on average 2.87%, and variable between individuals, but obviously not correlated with age (*Table 4*). Note that levels might have been a bit underestimated in Golgi material. The point will be addressed below.

For a more detailed analysis, we compared six cortical areas (primary sensory to limbic) in macaque using the same method, SMI-32/βIV-spectrin immunofluorescence. Antibody SMI-32 is directed against nonphosphorylated neurofilaments. It labels somata and dendrites of large type 1 pyramidal cells mainly of layers III and V, and much weaker the smaller pyramidal neurons, but not spiny stellates of layer IV and small pyramidal neurons of layer II (*García-Cabezas and Barbas, 2014*). βIV-spectrin is one of the most reproducible markers for the AIS. The following regions were assessed: visual cortex V1/operculum, auditory cortex A1 along the lower bank of the lateral fissure, somatosensory cortex S2 along the upper bank of the lateral fissure, cingulate cortex medial and lateral flank including areas 23 and 31, respectively, the upper and lower bank of the intraparietal sulcus, and dorsal cortex (premotor and parietal at more anterior levels). The intraparietal sulcus has on the upper bank area MIP (medial intraparietal area) involved in grasping, and on the lower bank the areas LIP (lateral intraparietal area) and VIP (ventral intraparietal area) involved in control of eye movements. Intraparietal neurons were retrogradely labeled from the premotor cortex injections. In three individuals, the percentages of AcD neurons varied between the six areas by about a factor of 2 (*Figure 4A*). We could not recognize a systematic difference in that one of the areas presented with substantially higher or lower percentages.

Furthermore, we compared primary visual area 17 to extrastriate visual areas in adult cats which received biocytin injections. Also here, individual percentages of AcD neurons varied from 11 to 20%; the interindividual variability was larger than the interareal variability. There was no recognizable difference between the areas analyzed (*Figure 4B*). Moreover, the values obtained in cat visual cortex matched those in ferret visual cortex (striate and extrastriate) (*Figure 4B*; *Table 2*).

As defined in the beginning, all neurons in which the axon origin was not unequivocally seen to emerge from a dendrite were scored as shared root cells, and for *Figure 3* and *Figure 4* were included in the group of cells with somatic axons. Yet, cells with the shared root configuration according to our criteria have been accepted in recent studies already as AcD neurons (*Thome et al., 2014*). The question was, how often does the shared root configuration according to our definition occur? We

**Table 2.** Proportion of pyramidal neurons with AcD: ungulate, carnivores.

| Species; cortical area; staining method; age; sex | Proportion of AcD cells [%]; n of cells assessed | By layers: supra %, infra % |
|---|---|---|
| **Kitten visual cortex gyral white matter; intracellular Lucifer yellow** | | |
| P1, P2 (n = 2, sex n.d.) | 37.93%, 58 cells | n.a. |
| P10, P11 (n = 2, sex n.d.) | 44.64%, 56 cells | n.a. |
| P12, P14 (n = 2, sex n.d.) | 47.37%, 19 cells | n.a. |
| *Average [%], total n of cells* | **43.31%, 133** cells | |
| **Kitten and adult cat visual cortex layer VI; intracellular Lucifer yellow** | | |
| P1, P5 (n = 2, sex n.d.) | 20.18%, 114 cells | n.a. |
| P11 (n = 2, sex n.d.) | 17.65%, 51 cells | n.a. |
| P19, P30 (n = 2, sex n.d.) | 14.89%, 47 cells | n.a. |
| P52, P60 (n = 2, sex n.d.) | 20.00%, 30 cells | n.a. |
| Adult (n = 2, sex n.d.) | 17.95%, 39 cells | n.a. |
| *Average [%], total n of cells* | **18.13%, 281** cells | |
| **Adult cat visual cortex; SMI-32/βIV-spectrin immunofluorescence** | | |
| Individual 1, 3 months, sex n.d. | 13.35%, 978 cells | 14.42%, 14.55% |
| Individual 2, adult, sex n.d. | 14.92%, 496 cells | n.d. |
| **Adult cat visual cortex; biocytin tracing** | | |
| Individual 3, adult, sex n.d. | 13.44%, 655 cells | 12.50%, 15.85% |
| Individual 4, adult, sex n.d. | 17.14%, 70 cells | 18.52%, 12.50% |
| Individual 5, adult, sex n.d. | 13.93%, 316 cells | 13.12%, 15.79% |
| Individual 6, adult, sex n.d. | 19.13%, 230 cells | 19.34%,18.37% |
| Individual 7, adult, sex n.d. | 12.64%, 174 cells | 11.49 %, 19.23% |
| *Average [%] individuals 1–7, total n of cells* | **14.94%, 2919** cells | **14.90%, 16.05%** |
| **Ferret visual cortex; biocytin tracing** | | |
| Individual 1, adult, female | 16.56%, 302 cells | 15.95%, 18.57% |
| Individual 2, adult, female | 14.66%, 191 cells | 15.39%, 13.11% |
| Individual 3, adult, female | 11.30%, 230 cells | 10.24%, 14.06% |
| Individual 4, adult, male | 13.07%, 329 cells | 13.45%, 12.26% |
| *Average [%], total n of cells* | **13.90%, 1052** cells | **13.76%, 14.50%** |
| **Pig dorsoparietal cortex; SMI-32/βIV-spectrin immunofluorescence** | | |
| 3 months, European wild boar, female | 20.11%, 189 cells | 22.04%, 19.05% |
| 5 months, domestic, sex n.d. | 20.99%, 181 cells | n.d. |
| *Average [%], total n of cells* | **20.55%, 370** cells | **22.04%, 19.05%** |

AcD, axon carrying dendrite; n.a., not applicable; n.d., not determined due to too weak staining.

**Table 3.** Proportion of pyramidal neurons with AcD: primates - macaque.

| Species; cortical area; staining method; age; sex | Proportion of AcD cells [%]; n of cells assessed | By layers: supra %, infra % |
|---|---|---|
| *Macaca mulatta* premotor cortex; biocytin tracing | | |
| Individual 1, 11 years, male | 4.93%, 954 cells | 2.79%, 11.38% |
| Individual 2, 7 years, male | 3.10%, 816 cells | 2.93%, 6.32% |
| Individual 3, 5 years, male | 4.26%, 423 cells | 3.23%, 6.25% |
| *Macaca mulatta* parietal and visual cortex; SMI-32/βIV-spectrin immunofluorescence | | |
| Individual 4, 5 years, male | 5.99%, 717 cells | 2.59%, 13.95% |
| Individual 5, 10 years, male | 6.75%, 681 cells | 2.51%, 12.72% |
| *Macaca fascicularis* parietal cortex; Golgi-Kopsch method | | |
| Individual 6, adult, female | 3.58%, 307 cells | 4.82%, 6.90% |
| Individual 7, adult, male | 3.07%, 228 cells | 1.57%, 5.67% |
| *Average [%] individuals 1–7, total n of cells* | **4.53%, 4126** cells | **2.92%, 9.03%** |
| *Macaca mulatta* visual cortex; biocytin tracing | | |
| Individual 8, P60, female | 4.51%, 377 cells | 1.28%, 5.88% |
| *Macaca mulatta* cingulate cortex; SMI-32/βIV-spectrin immunofluorescence | | |
| Individual 8, P60, female | 5.80%, 500 cells | 1.28%, 7.85% |
| *Macaca mulatta* premotor/M2 cortex; SMI-32/βIV-spectrin immunofluorescence | | |
| Individual 8, P60, female | 5.36%, 1249 cells | 1.34%, 9.05% |
| *Average [%] individual 8, total n of cells* | **5.22%, 2126** cells | **1.30%, 7.59%** |

AcD, axon carrying dendrite.

plotted the percentages of AcD neurons versus shared root neurons for rat, ferret, macaque, and human (*Figure 5A*). For macaque, we included the biocytin-stained material from premotor cortex. In macaque Individual 2 we could assess the contralateral cortex to determine the percentage of AcD and shared root cells of callosal projection neurons. Furthermore, in Individual 2, long-range projection neurons residing in the intraparietal sulcus were assessed, which in functional terms belong to the eye-hand coordination and grasping network. Furthermore, we determined the shared root configuration in all areas shown in *Figure 4A* visualized via immunofluorescence. Also included were the values obtained in Golgi-stained macaque and human cortex (see *Figure 5—source data 1*). In *Figure 5A*, the species cluster along the ordinate as already evident in *Figure 3B*. However, in *Figure 5A*, the species scatter widely along the abscissa. This suggested an absence of a systematic correlation between AcD and the shared root configuration.

Next, for rat, ferret, macaque, and human, we compared the percentages of AcD to the sum of AcD plus shared root (*Figure 5B*). If the shared root cells were considered as AcD cells, the proportions of AcD cells increase to some extent in all species analyzed. The interindividual variability of the shared root cells was at a factor of >10 (range of 0.46–5.5% in macaque), and statistics argued against any biologically significant difference between species.

Unexpectedly, a subtle difference was observed independently by two observers who analyzed the Golgi material (PW at Ruhr University Bochum, GM at University La Laguna). A total of 13 cases (2 macaque, 11 human individuals) had percentages of shared root cells higher than percentages of AcD cells, whereas in 22 of 25 individuals and/or cortical areas stained for biocytin and immunofluorescence the percentages of shared root cells were lower than the percentages of AcD cells (*Figure 5B*). Thus, the proportion of AcD neurons was slightly underrepresented in the Golgi material. Yet, also the biocytin material had larger proportions of shared root cells (*Figure 5B*). We therefore compared

**Table 4.** Proportion of pyramidal neurons with AcD: primates - human.

| Species; cortical area; staining method; age; sex | Proportion of AcD cells [%]; n of cells assessed | By layers: supra %, infra % |
|---|---|---|
| Human temporal lobe; Golgi-Cox method | | |
| Individual 1, 53 years, male | 2.56%, 646 cells | 1.45%, 4.40% |
| Individual 2, 56 years, male | 2.79%, 825 cells | 0.96%, 4.66% |
| Human auditory cortex; Golgi-Kopsch method | | |
| Individual 3, 63 years, male | 0.79%, 253 cells | 0.69%, 0.91% |
| Individual 4, 71 years, female | 1.92%, 677 cells | 0.80%, 2.58% |
| Individual 5, 75 years, male | 0.47%, 215 cells | 0.00%, 0.80% |
| Individual 6, 88 years, female | 0.71%, 140 cells | 0.00%, 1.28% |
| Individual 7, 56 years, female | 1.72%, 407 cells | 1.27%, 2.34% |
| Human prefrontal agranular cortex; Golgi-Kopsch method | | |
| Individual 8, 46 years, female | 2.02%, 247 cells | n.d. |
| Individual 9, 77 years, male | 2.59%, 424 cells | n.d. |
| Individual 10, 87 years, female | 1.53%, 653 cells | n.d. |
| Human visual cortex area 18 Golgi-Kopsch method | | |
| Individual 10, 87 years, female | 2.48%, 242 cells | 1.77%, 3.10% |
| *Average [%] individuals 1–10, total n of cells* | *1.96%, 4729* cells | *0.99%, 2.87%* |
| Human auditory cortex gyral white matter; Golgi-Kopsch method | | |
| Individual 3, 63 years, male | 8.69%, 115 cells, | n.a. |
| Individual 4, 71 years, female | 8.88%, 135 cells, | n.a. |
| Individual 5, 75 years, male | 8.70%, 69 cells, | n.a. |
| Individual 6, 88 years, female | 9.18%, 98 cells, | n.a. |
| *Average [%] individuals 3–6, total n of cells* | *8.86%, 417* cells | *n.a.* |

AcD, axon carrying dendrite.

immunofluorescence and biocytin of the macaque material (*Figure 5C*). Indeed, the biocytin material delivered significantly more shared root cells. The unequivocal AcD cells were equally well recognized by both methods. With the large data set of macaque Individual 2 we compared the two methods within one individual (*Figure 5D*). Again, similar to the Golgi material, the proportion of shared root cells was higher in the biocytin material than it was in the immunofluorescence material whereas unequivocal AcD cells were equally well detected with the two methods.

AcDs in rodent hippocampus are described as being functionally privileged. This may be mirrored by their spine density. Analysis of rat and ferret biocytin-stained pyramidal cells, however, revealed that neither the dendrites sharing a root with an axon nor the AcDs had spine densities differing systematically from the spine density of non-AcDs from the very same neuron (*Figure 6*).

## Developmental aspects

Kitten layer VI pyramidal cells (*Figure 3A*, *Table 2*) showed adult percentages of AcDs early postnatally. Many pyramidal cells were L-shaped or inverted-fusiform, with the axon emerging from one of the dominant dendrites (*Lübke and Albus, 1989*). In line with this, infant macaque cortex exhibited percentages of AcD cells comparable to adult cortex neurons labeled with biocytin, and again, AcD cells were more frequent in infragranular layers (*Figure 3A and B*, *Table 3*).

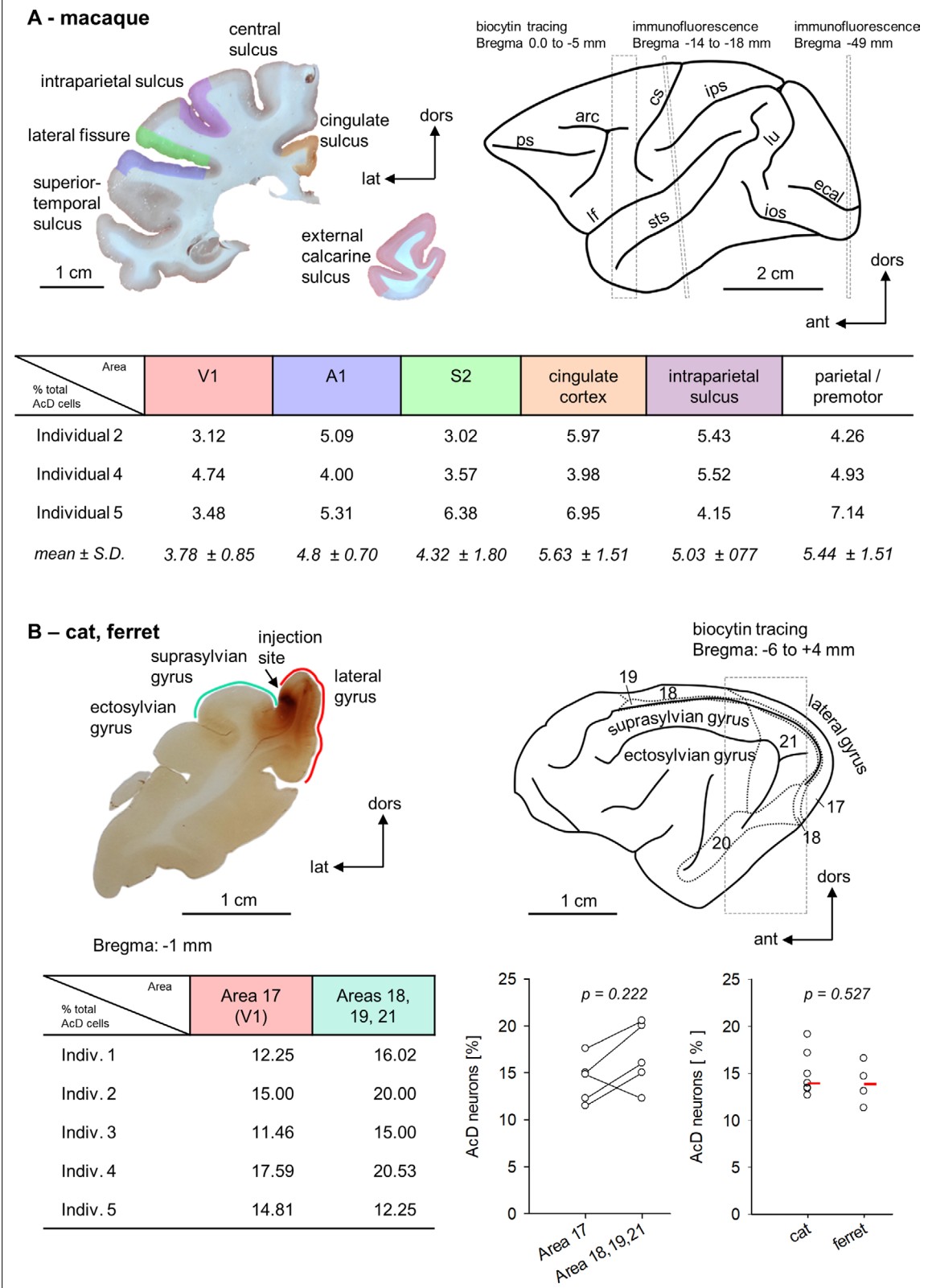

**Figure 4.** Within-species areal comparisons. (**A**) Upper left is a photomicrograph of one of the coronal sections stained for immunofluorescence. The regions of interest are color coded. Upper right is the macaque brain (after *Paxinos et al., 2009*). The dashed boxes and Bregma distances indicate where our assessments were made. The rostral box overlaps the premotor cortex harboring the biocytin injections. Note that the analysis was spanning several millimeters of cortex (see *Figure 4—source data 1*). The middle box corresponds to the level of the section shown to the left. It is slightly tilted

*Figure 4 continued on next page*

*Figure 4 continued*

with respect to the stereotaxis coordinates (***Paxinos et al., 2009***). The posterior box corresponds to a fairly caudal level of the visual cortex. The table summarizes the percentages of AcD neurons obtained in the six areas and three individuals and gives the mean of each area with standard deviation. Abbreviations: arc, arcuate sulcus; cgs, cingulate sulcus; cs, central sulcus; ecal, external calcarine sulcus; ios, inferior occipital sulcus; ips, intraparietal sulcus; lf, lateral fissure; lu, lunate sulcus; prs, principal sulcus; sts, superior temporal sulcus. (**B**) Upper left is a photomicrograph of one of the coronal sections of cat occipital cortex analyzed for biocytin-stained AcD neurons. The injection site in this case was near the area 17/18 border, some other cats had an additional injection into the suprasylvian gyrus (see ***Figure 4—source data 1***). Area 17 is along the medial flank, areas 18, 19, and 21 are in the lateral sulcus and on the suprasylvian gyrus. Upper right is the cat brain (after ***Reinoso-Suarez, 1961***) with the visual fields indicated. The table summarizes the percentages of AcD neurons obtained in area 17 and the extrastriate areas. The graph pairs the data points of the five cats. To the right, we compared cat (n = 7) to ferret (n = 4) visual cortex (striate and extrastriate). Every point is one individual, the red bar represents the median for each column. The p-values were determined with a Mann-Whitney rank sum test.

The online version of this article includes the following source data for figure 4:

**Source data 1.** Data and statistical analysis of experiments shown in ***Figure 4A and B***.

---

Unexpectedly, of the pyramidal cells in kitten white matter (***Wahle et al., 1994***), 43.31% had axons emerging from the major dendrite (***Table 2***). Even more striking, 8.86% of the interstitial pyramidal neurons of the adult human white matter (***Meyer et al., 1992***) displayed AcDs (***Table 4***), and on average an additional 13.23% of the interstitial pyramidal cells had axons emerging from a shared root.

## Interneurons

We also assessed the proportion of AcD of interneurons. In human Golgi material, interneurons were easily recognized by non-spiny slightly varicose dendrites, lack of polarity, and – if present – locally branching axons. The more grazile morphology with rather simple dendrites allowed a reliable detection of AcD cells. Examples of bitufted, Martinotti, and basket cells were Neurolucida reconstructed (***Figure 7A***). Not all cells had axons impregnated beyond the initial segment because some interneuronal axons are myelinated, yet, the AIS can be unequivocally distinguished from dendrites (***Jones, 1975***). Up to 30% of the interneurons (all types pooled) had an AcD (***Figure 7A***). This was in contrast to the rather small percentage of AcD pyramidal cells in human. Furthermore, Parvalbumin-positive neurons were analyzed in immunostained human material. About 22% had an AcD (***Figure 7A***). Parvalbumin is a marker for gamma-aminobutyric acid (GABA)-ergic fast-spiking basket and chandelier cells, whereas neuropeptides are enriched in non-fast-spiking interneurons with Somatostatin being a marker for this interneuron lineage, at least in a rodent. Somatostatin-positive neurons are GABA-ergic. In perinatal kitten occipital cortex, they start to appear in deep layer VI and gradually more cells differentiate in layer VI, V, and upper layers; many are bitufted cells and Martinotti cells with ascending axons (***Wahle, 1993***). Neuropeptide Y-positive cells of mainly layers VI and V of the gray matter are often small basket neurons and also belong to the GABA-ergic neurons. About 12% had an AcD. Neuropeptide Y-positive axonal loop cells of the cat subplate are a transient type of projection neuron (***Wahle and Meyer, 1987***), which was recently reported to not contain glutamate decarboxylase (***Ernst et al., 2018***). Only about 5% had an AcD, the vast majority of axonal loop cells had the axon originating from the soma. For both neuropeptide Y-positive cell types, this was constant through kitten postnatal development. Of the Somatostatin-positive neurons 45–50% had an AcD (***Figure 7B***). It seems as if the percentage of Somatostatin-positive AcD neurons would increase early after birth. However, this 'increase' rather reflected the differentiation of layer V/VI bitufted and Martinotti cells which begin to express Somatostatin more intensely. So, Somatostatin-positive AcD cells simply became easier to detect at higher numbers from P7-9 onward.

To summarize, we observed a substantial species difference with pyramidal AcD cells being more frequent in non-primates. Within species, we found clear laminar differences, with pyramidal AcD cells being rare in primate supragranular layers, and more frequent in deep layers, and in white matter (subplate/interstitial) neurons. Interneurons in human and kitten cortex presented with type-specific proportions of AcD which can be much higher than those of pyramidal neurons.

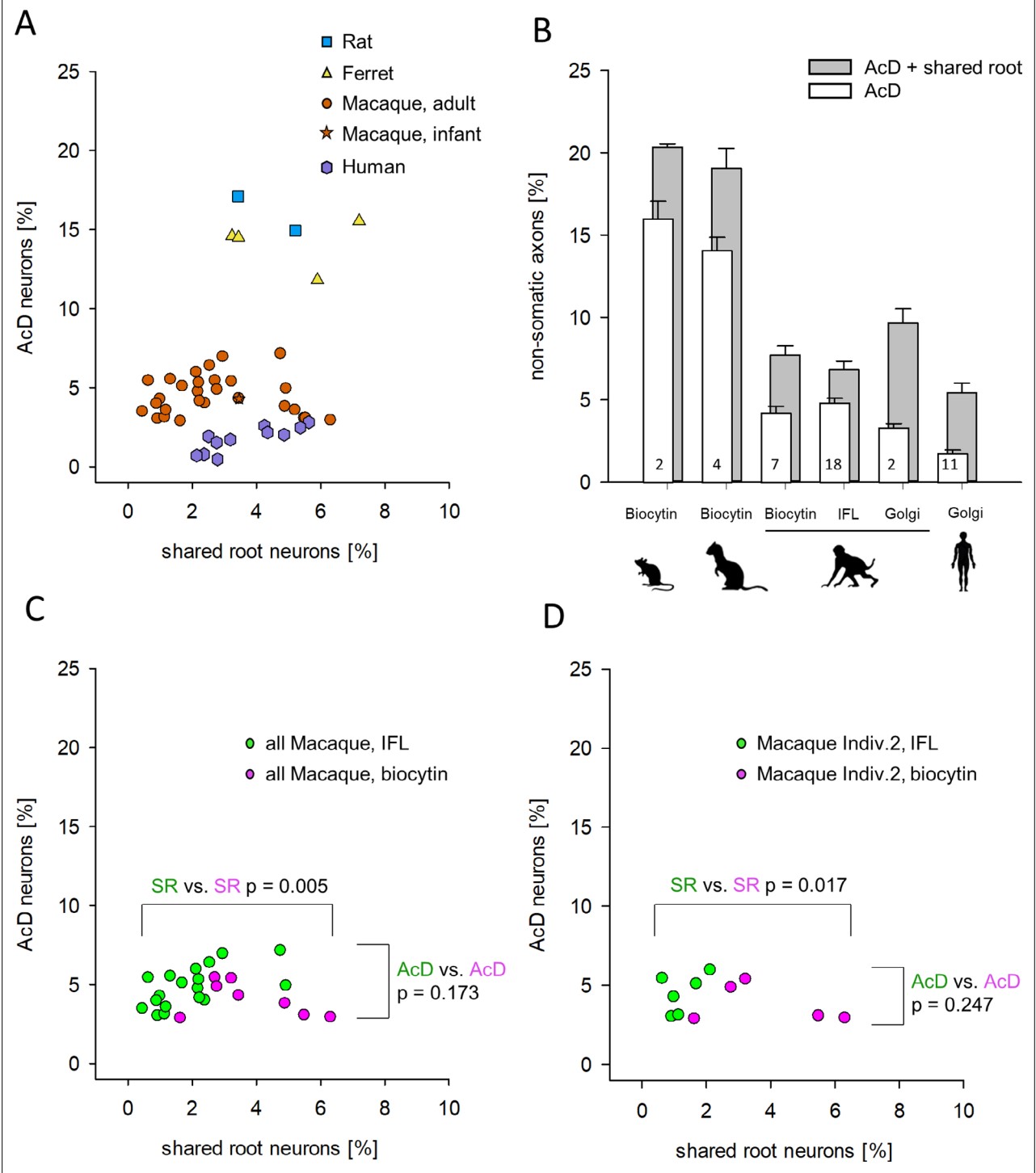

**Figure 5.** Proportion of axon carrying dendrite (AcD) cells versus shared root cells. (**A**) Data from rat (biocytin), ferret (biocytin), macaque (biocytin, immunofluorescence, Golgi), and human (Golgi). The species cluster along the ordinate as already seen in *Figure 3B*. The Mann-Whitney rank sum test of 'all non-primate' versus 'all macaque' proportions of AcD cells yielded p <0.001. However, the shared root values scatter considerably along the abscissa. Mann-Whitney rank sum test of 'all non-primate' versus 'all macaque' proportions of shared root cells yielded p=0.008. (**B**) The percentages of AcD were graphically compared to the sum of AcD and shared root. For macaque, data were separated by staining methods. Note that the Golgi method in macaque and in human yielded a higher proportion of shared root compared to biocytin and immunofluorescence. Numbers in the bars represent the sample size (individuals and/or cortical areas). (**C**) Comparison of biocytin and immunofluorescence staining in macaque. (**D**) Comparison of biocytin and immunofluorescence staining within just one individual macaque. Note in C, D that AcD cells are detected equally well with both

*Figure 5 continued on next page*

*Figure 5 continued*

methods whereas the biocytin staining yielded higher numbers of shared root cells (SR). In C, D, colors indicate the comparisons, and the p-values were determined with a Mann-Whitney rank sum test. IFL, immunofluorescence.

The online version of this article includes the following source data for figure 5:

**Source data 1.** Data and statistical analysis of experiments shown in *Figure 5A and B*.

## Discussion

A majority of human gray matter pyramidal neurons have axons arising from the soma. In this aspect, in particular supragranular neurons of primates differ from those of non-primates. We found an inter-individual variability of AcD cells at about a factor of 2, and despite our high cells numbers a sampling bias cannot be completely excluded. We could not find areal differences in macaques and in cats. Also, the data of human visual, auditory, temporal, and prefrontal cortex did not argue for areal differences. Basal dendritic trees of layer III pyramidal cells in human visual cortex are largest at birth whereas those in temporal cortical areas continue to increase in complexity during the first postnatal years (for review *Elston and Fujita, 2014*). This suggested that the sparcity of the AcD phenotype in human in particular in supragranular layers is not dependent on postnatal changes of dendritic complexity. An additional fraction of neurons have axons which share a root with a basal dendrite. Electron microscopy has demonstrated the mixed nature of the shared root which displays a dendritic fine structure but also contains the fasciculated microtubuli characteristic for the axon hillock. The latter is less distinct when the axon emerges from a dendrite in that the dense undercoating typical for the initial segment starts immediately after the point of divergence (*Peters et al., 1968*). This begs the question of how an axon can emerge from a dendrite? Cortical pyramidal neurons migrate radially upward while their axons emerge from the basal somatic pole and already during soma migration descend into the white matter. After the neurons have reached their laminar destination the leading process transforms into the apical dendrite, and basal dendrites begin to sprout. It remains

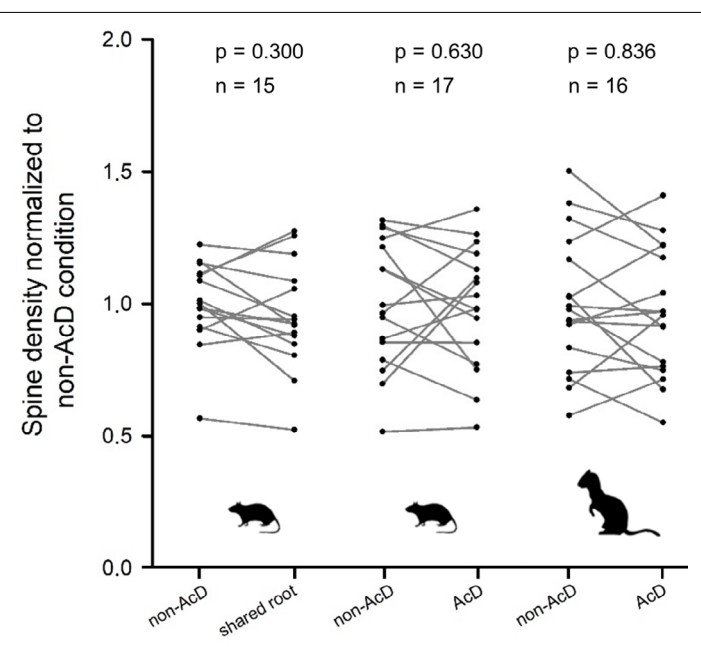

**Figure 6.** Spine density did not systematically differ between regular dendrites (non-axon carrying dendrites [non-AcDs]), dendrites sharing a root with a neighboring axon, and AcDs. Data from adult rat and ferret biocytin material, values from each cell are connected by a line. For normalization, the average of the non-AcD has been set to 1, and all values were expressed relative to this. Mann-Whitney rank sum test p-values and the sample size are reported above each plot.

The online version of this article includes the following source data for figure 6:

**Source data 1.** Data and statistical analysis of experiments shown in *Figure 6*.

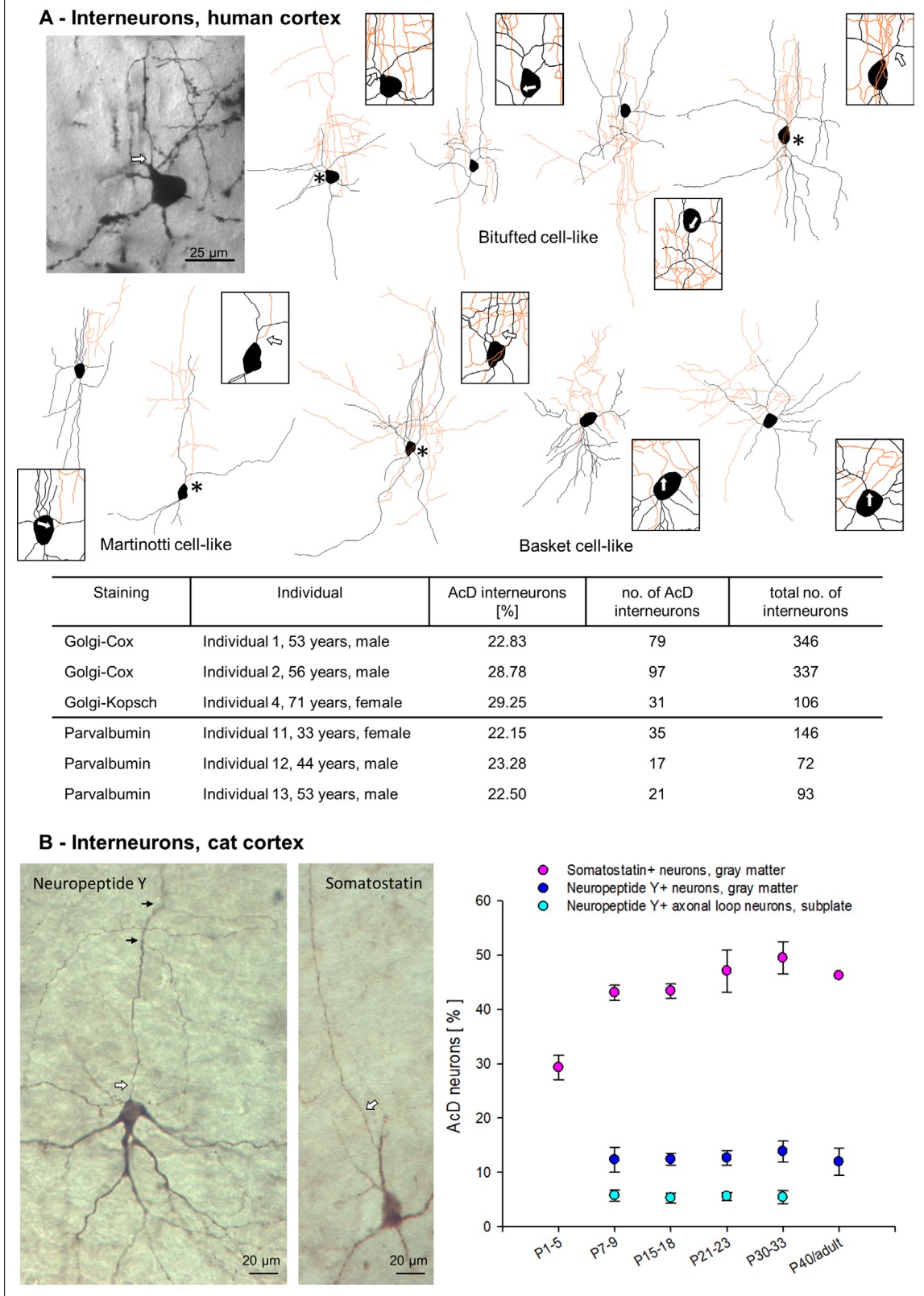

**Figure 7.** Axon carrying dendrite (AcD) interneurons in human and cat cortex. (**A**) Photomicrograph of a representative Golgi-impregnated bitufted neuron with arcade-like initial axon from supragranular layers next to its reconstruction, followed by three further examples of bitufted, Martinotti (2), and basket cells (3). Axons in orange, somata and dendrites in black. Asterisks mark AcD neurons, boxes with arrows show the axon origin at higher magnification. In the table, the percentage of Golgi-impregnated AcD interneurons is reported for Individuals 1, 2, 4, followed by the percentages of

*Figure 7 continued on next page*

*Figure 7 continued*

Parvalbumin-positive AcD neurons of Individuals 11–13. (**B**) Photomicrograph of a layer VI neuropeptide Y-positive neuron with somatic axon, and a layer V Somatostatin-positive AcD neuron. Axons marked by white arrows, small black arrows mark collaterals. The graph shows percentages of AcD interneuron subsets at the ages indicated in developing cat occipital cortex (see *Figure 7—source data 1* for sample size).

The online version of this article includes the following source data for figure 7:

**Source data 1.** Data and statistical analysis of experiments shown in *Figure 7A and B*.

to be shown if, during basal dendritogenesis, the axon hillock becomes passively displaced from the soma onto an outgrowing dendrite. However, the argument does not explain why the proportion of AcD neurons is much higher in hippocampus although numbers published with intracellular labeling methods recently in rodent CA1 neurons vary from 20 (*Benavides-Piccione et al., 2020*) to about 50% (*Thome et al., 2014*).

Furthermore, at this moment it is not clear if the axonal origin is always firmly anchored or can drift along the plasma membrane, for instance, by mechanical influences. It is known that the AIS is a regulated microdomain (*Jamann et al., 2018*) which undergoes activity-dependent shifts in length and in position. So, could the axon hillock actively 'translocate' or become passively displaced from the somatic to a proximal dendritic membrane? Dendrites are dynamic structures and although imaging studies in mouse have reported fairly stable basal dendrites of supragranular pyramidal neurons during development (*Trachtenberg et al., 2002*), there are also reports on dynamic changes elicited by environmental enrichment, activity, or disease (for review *Hickmott and Ethell, 2006*; *Elston and Fujita, 2014*).

Domestic pig and wild boar had similar proportions suggesting that domestication has no influence. Kitten and infant macaque data suggest that adult proportions of AcD neurons are already present at early ages, and the three assessments in infant macaque are within the macaque cluster (*Figure 3B*). In macaque, ontogenetically older infragranular pyramidal cells display more AcDs than later generated supragranular neurons, and the same was observed in our human material. Neurons of the white matter seem to be a special case. In cat, the inverted pyramidal neurons represent a subset of subplate cells. They reside at strategic positions to monitor incoming inputs and may quickly relay that information to the overlying gray matter via axons ascending into the gray matter including layer IV (*Friauf et al., 1990*). Given that subcortical afferents and white-to-gray matter projections match in topography (reviewed by *Molnár et al., 2020*), a synaptic double-hit scenario has been postulated with geniculocortical afferents trying to strengthen synapses onto layer IV spiny stellates, and with excitatory subplate afferents transiently acting as 'helper synapse' and instructor for the developing thalamocortical connectivity (reviewed by *Molnár et al., 2020*). With regard to the functional concept of AcD neurons (*Thome et al., 2014*; *Hamada et al., 2016*; *Kole and Brette, 2018*), our findings suggest that action potential firing abilities bypassing somatic integration and somatic inhibition are advantageous during development of thalamocortical wiring. In adult human white matter, pyramidal interstitial cells may differ from the transient subplate cells of non-primate cortex (*Meyer et al., 1992*; *Suarez-Sola et al., 2009*; *Sedmak and Judaš, 2021*). Yet, a function of quickly relaying incoming afferent information up to the gray matter is also conceivable here, and this might narrow the time window of synaptic integration enabling plasticity or help to activate inhibitory interneurons.

Whether neurons with axons sharing a common root with a dendrite should be regarded as AcD neurons is a matter of debate. From the morphological perspective, we assigned AcD in a very conservative manner. All neurons in which the axon origin was not unequivocally arising from a dendrite or seemed to share a common root with a dendrite were included in the group with somatic axons. Yet, in recent studies cells with the shared root configuration have been considered AcD neurons, also using immunofluorescence (*Hamada et al., 2016*; *Thome et al., 2014*). As expected, when plotting the sum of AcD plus shared root for the various staining methods, values for all species were increasing. However, the non-primate-to-macaque difference can be easily seen. For instance, our summed values from adult rat visual cortex sampled across all layers come closer to proportions reported for layer V neurons by *Hamada et al., 2016*; about 28% in adult Wistar rat somatosensory cortex. Of note, however, *Hamada et al., 2016* reported on neurons which by our criteria would not be AcD cells; their criterion for inclusion has been the distance of the spectrin/ankyrin G-labeled AIS to the soma irrespective of whether the axon emerges from a shared root or unequivocally from a dendrite.

Together, considering a fraction of shared root will be tolerable, at least in non-primate mammals with their substantial numbers of unequivocal AcD neurons.

The human Golgi material yielded the lowest values of AcD and of AcD plus shared root cells (*Figure 5B*) and the lowest proportion in supragranular layers (*Figure 3B*). We did not run statistical comparisons with our human data for the following reason. After analyzing more and more individuals and/or brain areas it became evident that the Golgi methods yielded a lower proportion of AcD neurons and the higher proportion of shared root cells. In line with this, also the biocytin material yielded higher proportions of shared root cells. A parsimonious explanation may be as follows. The Golgi reaction product is a chromate precipitate deposited at the plasma membrane. The pitch-black reaction product, the thickness of the tissue sections, on top of the complexity of basal dendrites in primate (*Hendry and Jones, 1983*) and even more so in human (*Mohan et al., 2015*; review by *Goriounova and Mansvelder, 2019*) can make it difficult to determine if an axon emerges from a soma, or from a shared root, or already from a very proximal dendritic trunk. The same accounts for the black biocytin reaction product (see *Figure 2* and *Figure 2—figure supplements 1 and 2*) although the section thickness here was thinner. An additional argument comes from the axon itself. Axons originating from dendrites are thinner and have less prominent hillocks (*Peters et al., 1968*; *Mendizabal-Zubiaga et al., 2007*; *Benavides-Piccione et al., 2020*). With dark reaction products it was difficult to precisely determine where exactly a thin process lacking a clear hillock arises from a large dendritic root. This way we counted somewhat higher percentages of shared root and somewhat lower percentages of AcD in the Golgi Cox and Golgi-Kopsch material. By contrast, the intracellular staining of much thinner sections such as the 20–50 µm thick sections of the biocytin and immunofluorescence material allowed to visualize structures at better optical resolution. In particular, the confocal analysis allowed to walk micrometer-by-micrometer through the optical stack to decide 'pro AcD' or 'pro shared root' for each case in question arguing that the optical resolution was the crucial parameter. Nevertheless, biocytin staining was equal to the immunofluorescence in detecting clear-cut AcD, but was inferior to immunofluorescence and confocal analysis when it comes to deciding on shared root. It should be noted that the frequently used SMI-32 staining method may also have a certain bias in that it stains preferentially type 1 pyramidal neurons (*Molnár and Cheung, 2006*). Future studies are needed before a final conclusion on the areal and laminar proportion of human pyramidal AcD neurons can be made, and for a species comparison intracellular staining methods should be applied as recently done for CA1 pyramidal cells (*Benavides-Piccione et al., 2020*).

Pyramidal cell AcDs in isocortex and allocortex are basal dendrites. We found less than 10 axons in rat, ferret, and macaque emerging from an apical dendrite of a classical upright pyramidal cell of layers II–V. Pyramidal cells of layer VI can be L-shaped or fusiform-bipolar with two major dendrites, or inverted, in rodent as well as in primate (*Hendry and Jones, 1983*). In human, the large-sized Von Economo neurons in cingulate and other cortices have been reported to regularly have an axon from a thick descending basal dendrite which in addition often shares a common root with a secondary dendrite (*Banovac et al., 2021*). A study comparing human and mouse hippocampal CA1 pyramidal cells with intracellular injections reported that axons may arise from basal dendrites. The proportions are 40% AcD cells in human and 20% AcD cells in mouse (*Benavides-Piccione et al., 2020*). The latter proportion differs markedly from the 52% AcD neurons visualized by DsRed expression in mouse CA1 neurons, and the 47% AcD neurons visualized via intracellular injection in Wistar rat CA1 neurons reported by *Thome et al., 2014*. Electron microscopy has revealed that in rat cortex the AIS of axons originating from one of these major dendrites of inverted pyramidal cells are thinner (*Peters et al., 1968*; *Mendizabal-Zubiaga et al., 2007*; *Benavides-Piccione et al., 2020*), and the initial segment is shorter and less innervated by symmetric synapses than AIS of axons arising from the soma (*Mendizabal-Zubiaga et al., 2007*). In cat visual cortex, inverted-fusiform pyramidal neurons of layer VI serve corticocortical, but not corticothalamic projections; for instance, the feedback projection to area 17 from the suprasylvian sulcus (*Einstein, 1996*), an area involved in motion detection, processing of optical flow, and pupillary constriction. With regard to the functional concept of AcD cells, the kinetics of intra- and interareal information processing may have so far unrecognized facets.

Local axon GABA-ergic cortical interneurons often have axons emerging from dendrites in rodent as well as in monkey (*Jones, 1975*) and human (*Kisvárday et al., 1990*). Our data confirm earlier observations in vivo (*Wahle and Meyer, 1987*; *Meyer and Wahle, 1988*; *Wahle, 1993*) and in vitro (*Höfflin et al., 2017*) in that the frequency of AcD is cell type specific. About half of the bitufted and

Martinotti neurons in cat cortex had an AcD whereas most Parvalbumin-positive neurons, in particular, large basket cells in the human cortex had somatic axons regardless of laminar position. Intriguingly, cat subplate axonal loop cells turned out to be lowest with just about 5% AcD cells. This was in contrast to >40% AcD subplate pyramidal cells present at the very same ages in the very same compartment, with both types being co-generated from the cortical ventricular zone early during corticogenesis. Also intriguingly, about one-third of the interneurons of human cortex had an AcD. Our sample represents a mixture of types because the axons were stained only initially, and it was not possible to separate by type, as numbers were too small for this. Interestingly, however, the proportion of AcD interneurons in human was fairly close to an average proportion of AcD interneurons in cat cortex, whereas the proportion AcD pyramidal cells in human was substantially lower compared to pyramidal cells of cat and other non-primate mammals. Why interneurons do not seem to follow the primate trend toward less AcD cells remains to be unraveled.

Our data add to the view that human cortical pyramidal neurons differ in important aspects from those of non-primates (*Elston et al., 2011*; *Elston and Fujita, 2014*; *Defelipe, 2011*; *Beaulieu-Laroche et al., 2018*; *Gidon et al., 2020*; *Rich et al., 2021*). For instance, human supragranular pyramidal neurons have highly complex basal dendrites (*Hendry and Jones, 1983*), each being a unique computational unit (reviewed by *Goriounova and Mansvelder, 2019*). Furthermore, layer II/III human pyramidal cell dendrites have unique membrane properties (*Eyal et al., 2016*) and are more excitable than those of rat (*Beaulieu-Laroche et al., 2018*). Another feature is the unique design of the human cortical excitatory synapses having pools of synaptic vesicles, release sites, and active zones that are much larger compared to those in rodents (*Molnár et al., 2016*; *Yakoubi et al., 2019*). Large and efficient presynapses and more excitable dendrites may reliably depolarize the target cell's somatodendritic compartment such that electrical dendroaxonic short circuits might become obsolete. We propose, from non-primate to primate isocortical pyramidal neurons, an evolutionary trend toward inputs that are conventionally integrated within the somatodendritic compartment and can be precisely modulated by inhibition to generate an optimally tuned cellular and, finally, behavioral output.

# Materials and methods

## Key resources table

| Reagent type (species) or resource | Designation | Source or reference | Identifiers | Additional information |
|---|---|---|---|---|
| Antibody | anti-βIV-spectrin (Rabbit polyclonal) | *Höfflin et al., 2017* | self-made | IF (1:500) |
| Antibody | anti-SMI-32 unphosphorylated neurofilaments (Mouse polyclonal) | Covance, Muenster, Germany | Cat # SMI-32r, RRID: AB_2315331 | IF (1:1000) |
| Antibody | anti-NeuN (Mouse monoclonal) | Merck (Millipore), Darmstadt, Germany | Cat # MAB377, RRID: AB_2298772 | IF (1: 1000) |
| Antibody | anti-Parvalbumin, (Rabbit recombinant) | SWANT, Marly Switzerland | Code No. PV 27 | IHC (1:5000) |
| Antibody | donkey anti-rabbit (Alexa-488 polyclonal) | Thermo Scientific, Waltham MA, USA | RRID: AB_2687506 | IF (1:1000) |
| Antibody | goat anti-rabbit (biotinylated polyclonal) | Dako A/S, Glostrup, Denmark | RRID: AB_2313609 | IHC (1:1000) |
| Antibody | goat, anti-mouse (Alexa-568 polyclonal) | Invitrogen, Carlsbad, CA, USA | RRID: AB_2534013 | IF (1:1000) |
| Antibody | sheep anti-mouse (biotinylated polyclonal) | GE Healthcare Life Sciences, Braunschweig Germany, | Amersham Cat # RPN1001 | IHC (1:200) |
| Chemical compound, drug | Streptavidin Alexa-488 | Thermo Scientific, Waltham MA, USA | Cat #S11223 | (1:1000) |
| Chemical compound, drug | ABC Elite horseradish peroxidase | Vector Labs Inc, Burlingame, CA, USA, | RRID: AB_2336827 | (1:250) |
| Other | DAPI stain | Invitrogen | D1306 | (1 μg/mL) |
| Software, algorithm | SigmaStat 12.3 | Systat Software GmbH | Frankfurt am Main, Germany | |

## Animals

The data presented here were compiled by tissue sharing (immunohistochemistry) and from tissue that had originally been processed for unrelated projects, i.e., no additional animals were sacrificed specifically for this study.

## Biocytin injections

Two adult male Long-Evans pigmented rats (*Table 1*) received local biocytin injections into areas 17 and 18 in the course of teaching experiments done in the 1990s demonstrating surgery, tracer injections, and biocytin histology. The animals were from the in-house breeding facility. The histological material has been used for decades to train neuroanatomy course students at the Department of Zoology and Neurobiology. Four adult pigmented ferrets (*Mustela putorius furo*, *Table 2*) received biotin dextrane amine (BDA) injections into the motion-sensitive posterior suprasylvian area (*Philipp et al., 2006*, *Kalberlah et al., 2009*). Five adult cats (*Table 2*) received biocytin injections into visual cortex at around the border of area 17 to area 18 (Distler and Hoffmann, unpublished). After a survival time of 6–13 days, the animals were sacrificed and processed as described for the macaque cases. Three male adult macaques (*Macaca mulatta*; *Table 3*) received tracer injections (15–20% BDA MW 3000) into dorsal premotor cortex (*Distler and Hoffmann, 2015*). After a survival period of 14–17 days, the animals were sacrificed with an overdose of pentobarbital and perfused through the heart with 0.9% NaCl and 1% procaine hydrochloride followed by paraformaldehyde-lysine-periodate containing 4% paraformaldehyde in 0.1 M phosphate buffer pH 7.4. Coronal 50 µm thick frozen sections were cut on a microtome and processed for biocytin histochemistry with the avidin-biotin method (ABC Elite) with diaminobenzidine as chromogen which, in most cases, was enhanced with ammonium nickel sulfate (*Distler and Hoffmann, 2015*). A P60 infant macaque (*Table 3*) received a biocytin injection into visual cortex and has been processed as above (*Distler and Hoffmann, 2011*).

## Intracellular Lucifer yellow injections

The cat material was from a study on development of area 17 layer VI pyramidal cell dendrites (*Lübke and Albus, 1989*). Briefly, Lucifer yellow was iontophoretically injected into the somata in fixed vibratome slices of 100–150 µm thickness followed by UV-light photoconversion in the presence of diaminobenzidine toward a solid dark-brown reaction product (*Table 2*). Furthermore, we assessed neurons in the white matter of developing cat visual cortex (*Table 2*) prelabeled with the antibody 'subplate-1' followed by Lucifer yellow injection and photoconversion (*Wahle et al., 1994*).

## Immunofluorescence

Mouse material (*Table 1*) was collected as part of the ongoing dissertations of Nadja Lehmann and Susanna Weber, Institute of Neuroanatomy, Medical Faculty Mannheim, Heidelberg University, supervised by Prof. Maren Engelhardt. Sections were processed as described previously (*Jamann et al., 2021*). The intrinsic EGFP signal was combined with βIV-spectrin immunostaining. Adult cat material for immunostaining was from studies on development of visual cortex interneurons (*Wahle and Meyer, 1987*; *Meyer and Wahle, 1988*). Cryoprotected slabs of these brains had been stored since then embedded in TissueTek at –80°C. The 5-month pig material was obtained from the Institutes of Physiology and Anatomy, Medical Faculty, University Mannheim (donated by Prof. Martin Schmelz). The P90 European wild boar material was from current studies (*Ernst et al., 2018*; *Sobierajski et al., 2022*; *Table 2*). Adult macaque and P60 infant macaque material not used for immediate histological assessment had been stored after fixation and glycerol infiltration in isopentane at –80°C. From such spared blocks, 50-µm cryostat sections were cut for immunostaining (*Table 3*). Sections were pretreated with 3% $H_2O_2$ in TBS for 30 min, rinsed, incubated for 1 hr in 0.5% Triton in TBS, blocked in 5% horse serum in TBS for 2 hr followed by incubation in mouse anti-SMI-32 to stain somata and dendrites of subsets of pyramidal cells, and rabbit anti-βIV-spectrin (*Höfflin et al., 2017*) to stain the AIS. We could do only one pig and one cat for the laminar analysis because the immunofluorescence did not deliver sufficient basal dendritic SMI-32 labeling of supragranular neurons in the second available individual. Thus, for these two cases no reliable laminar data could be obtained. Mouse anti-NeuN staining of adjacent sections helped to identify the layers in the biocytin material. DAPI counterstaining helped to identify layers of the immunofluorescence sections. After 48 hr incubation at 8°C sections were rinsed, incubated in fluorescent secondaries including DAPI to label nuclei,

and coverslipped for confocal analysis. Formalin-fixed human material donated to the Department of Anatomy of the University of La Laguna (see below, Golgi-Kopsch method) was cryosectioned at 80 µm thickness and immunoperoxidase stained for parvalbumin to determine AcD basket and chandelier cells (Individuals 11–13 in *Figure 7A*). The material was prepared as part of the dissertation of Maria Luisa Suarez-Sola (University La Laguna, Spain, 1996) under the supervision of Prof. Gundela Meyer; the material served to illustrate the publication *Suarez-Sola et al., 2009*. Stained sections were reassessed for the present study.

## Golgi impregnation

The Golgi-Cox impregnations were done with so-called access tissue removed during transcortical amygdalo-hippocampectomy from two adult patients who suffered from temporal lobe epilepsy (Individuals 1 and 2 in *Table 4*). All experimental procedures were approved by the Ethical Committees as reported (*Schmuhl-Giesen et al., 2021*; *Yakoubi et al., 2019*). These and other previous studies *Mohan et al., 2015* have demonstrated that the access tissue is normal because it is located far from the epileptic focus. Biopsy tissue was processed using the Hito Golgi-Cox Optim-Stain kit (Hitobiotec Corp) as described (*Schmuhl-Giesen et al., 2021*; *Yakoubi et al., 2019*). Coronal sections (quality as shown by *Schmuhl-Giesen et al., 2021* in their Figure 1—figure supplement 1) were analyzed for AcD pyramidal neurons in supragranular (I–IV) and infragranular (V–VI) layers. Furthermore, interneurons with smooth dendrites were assessed from this material. Of selected neurons the initial axon and dendrites were 3D reconstructed with the Neurolucida (MicroBrightField Inc, Williston, VT, United States) at 1000× magnification.

The Golgi-Kopsch impregnations of macaque cortex (*Table 3*) were done on spare tissue from experiments done by Prof. Dr. Barry B. Lee (*Lee et al., 1983*). The sections had been used as reference material in the Dept. of Anatomy, University of La Laguna, Tenerife, Spain. The Golgi-Kopsch impregnations of human auditory and agranular prefrontal cortex (Individuals 3–9 in *Table 4*) were processed decades ago (*Meyer, 1987*; *Meyer et al., 1989*; *Meyer et al., 1992*). The brains were from notarized donations to the Department of Anatomy of the University of La Laguna for teaching of medical students and for research. Donors had no neurological disorders. After death, the bodies were transferred to the Department and perfused with formalin. The brains were extracted, stored in formalin, and small selected blocks were processed using a variant of the Golgi-Kopsch method. Tissue blocks were immersed in a solution of 3.5% potassium dichromate, 1% chloral hydrate, and 3% formalin in distilled water for 5 days, followed by immersion for 2 days in 0.75% silver nitrate. Blocks containing the auditory cortex (Heschl's gyrus), ventral agranular prefrontal cortex, and visual area 18 were cut by hand with a razor blade, dehydrated, and mounted in Epon. For the assessment of AcD cells in the white matter, the border between gray and white matter was traced (*Meyer et al., 1992*). We avoided this zone and took as orientation the dense aggregations of astrocytes in the white matter and the linear arrangement of blood vessels. As shown before (*Meyer et al., 1992*), interstitial pyramidal cells have a variety of shapes, from elongated bipolar to multipolar, but carry dendritic spines in contrast to non-pyramidal interstitial cells.

## Analysis and assignment of AcD

We scored all pyramidal cells with sufficiently well-stained basal and apical dendrites that had a recognizable axon. We analyzed fields of view where the labeled pyramidal cells are fairly perpendicularly oriented such that the apical dendritic trunk and the descending axon could be clearly seen. In the biocytin, Lucifer yellow, and Golgi cases, the axon could be clearly distinguished from sometimes equally thin descending dendrites because the latter had spines. Axons often had a clear axon hillock, which is more prominent in primate > carnivore > rodent, although this was less prominent for axons emerging from dendrites. Descending primary axons often gave rise to thinner collaterals. Note the complementary nature of the methods: Golgi impregnation labels neurons in all layers, not selecting for particular types of pyramidal cells, and also yielding cells of layer II which are SMI-32-negative. The Golgi-Cox method with the optimized kit yields a somewhat higher density of neurons than the Golgi-Kopsch method. Yet, the various Golgi methods have been reported to deliver very similar results (*Banovac et al., 2021*). Intracortical biocytin/BDA injections labeled preferentially neurons with horizontal projections in layers II/III, and neurons of infragranular layers closer to the injection site. SMI-32/βIV-spectrin labeling is strongest in large pyramidal cells of layer III and also in infragranular layers, in

particular of layer V, and weaker in layer VI as demonstrated (*Paxinos et al., 2009*). This way, the two methods yielded data preferentially for type 1 pyramidal neurons. The areal comparison in macaque is reported in *Figure 4A*. The fields were identified according to *Paxinos et al., 2009* and *Lewis and Van Essen, 2000*. The areal comparison in cat is reported in *Figure 4B*. The fields were identified according to *Reinoso-Suarez, 1961*.

All neurons fulfilling the criteria were sampled by five observers trained on the AcD criteria (ME, Linz; GM, La Laguna; PW together with IG or ES, Bochum, mostly by '4-eyes'). For light microscopy, neurons were viewed and scored with 40× and 63× objectives. All sections available of the tracing material were assessed and described in the source data, and in part more than once. In a first assessment, we went perpendicular to obtain AcD cells in all layers. These data are largely included in *Figure 3A*. However, when analyzing the macaque material we got the impression of less AcD neurons in supragranular layers. Therefore, we assessed the macaque and non-primate biocytin material a second time, now in a surface-parallel manner. Furthermore, for the laminar analysis, we had to obtain larger numbers of neurons to minimize any sampling error. With most of the human Golgi material, AcD cells were determined in a laminar fashion from the beginning, and since we found so few AcD cells we also scored the shared root configuration from the beginning. Subsequently, to obtain the shared root from the biocytin material for a species comparison, we reassesed the animal material a third time, again in a perpendicular manner albeit in fewer sections. Cell numbers and the histological basis for every figure are given in the source data, note that total cell numbers vary but the percentage of AcD cells obtained was in all cases close to the first count of the individual and within the range for each species.

For SMI-32/βIV-spectrin and Thy-1/βIV-spectrin fluorescence, images and the tile scan were done with a Leica TSC SP5 confocal microscope (40× and 10× objective resp., with 1.1 NA, 1024 × 1024 px). For SMI-32/βIV-spectrin fluorescence, the areas were imaged by taking confocal stacks at regular distances in supragranular and infragranular layers to equal proportions. All stacks (numbers are given in the source data files) were quantitatively assessed, no selection was made. We aimed to obtain large numbers of neurons in order to avoid or at least reduce any sampling bias as much as possible for the laminar analysis, and in particular for the macaque tissue. Therefore, all neurons (AcD, shared root, somatic) with sufficient staining of the initial dendrites and a βIV-spectrin-labeled AIS were manually marked in the confocal stacks using the '3D-environment'-function of Neurolucida 360 similar to *Figure 2—videos 1–3* exported from the Leica program. For the photomicrographs presented, global whole picture contrast, brightness, color intensity, and saturation settings were adjusted with Adobe Photoshop. Scale bars were generated with ImageJ (MacBiophotonics) and inserted with Adobe Photoshop (CS6 Extended, Version 13.0 × 64).

The assignment of AcD was done in a very conservative manner following *Peters et al., 1968* (see their Figure 1 with cell A presenting a shared root, cell B a somatic axon; cells C, D are AcD cells). Thus, we accepted as AcD cells only neurons in which the axon arose with recognizable distance of at least the width of the axon hillock to the soma, or emerged at such an angle that a vector through the axon hillock will not project into the soma, but will bypass the soma tangentially. Sometimes the axon and a dendrite emerged so close to each other or from a shared root (in X/Y but also Z level) such that the optical plane did not allow to make a clear decision. We included shared root cells in the group of 'somatic axon cells', unless otherwise noted/analyzed (see *Figure 5*). In particular, the white matter pyramidal neurons of the human brain and of the cat brain were difficult due to their elongated shape and the somata tapering into the major dendrites (*Meyer et al., 1992*). Therefore, we strictly aimed for the clear-cut cases.

## Spine analysis

To elucidate if the privileged AcD has a higher spine density than non-AcD, spines were plotted with the Neurolucida at 1000× magnification from biocytin-labeled neurons of rat and ferret cortex from primary and secondary basal dendrites starting minimum 50 µm away from the soma. On average we were able to reconstruct 170 µm/neuron in rat and 145 µm/neuron in ferret. The number of spines per 100 µm dendritic length was computed, and the value for the AcD was paired to the average value of the basal non-AcD of every neuron. Yet, the number of measurable neurons was limited for the following reasons. First, neurons had to be well backfilled with the tracer. Second, neurons had to have an appreciable length of the AcD plus a minimum of one basal non-AcD in the 50 µm thin

sections. Third, these dendrites had to display branch orders of 2–4, because the proximally thicker stems are not suitable for spine analysis and often void of spines (*Hübener et al., 1990*). Fourth, only solitary cells residing not too close to the injection site with its high background could be analyzed. Spine densities varied in our data set. Technically, the degree of biocytin labeling expectedly varied with the strength of the connection to the injection site. Biologically, pyramidal cell type-specific spine densities are known to vary up to an almost spine-free state, e.g., in Meynert cells (*Hübener et al., 1990*). To collect a sufficient sample size, we included moderately biocytin-backfilled cells, although they tended to present with a lower spine density. Moreover, most counts were taken from branch order 2–4 segments which may have less spines than terminal segments. Our density average in rat matched values reported for nonterminal segments of Golgi-stained near-adult hooded rat visual cortex supragranular pyramidal cells (*Juraska, 1982*). Our ferret spine values were lower compared to earlier reports (*Clemo and Meredith, 2012*) presumably for the reasons mentioned above. However, this would not compromise our finding because we compared only dendrites within the individual neurons. Would there be some systematic change of the spine density between the AcD and the non-AcD of each cell, the difference should manifest irrespective of the individual staining intensity.

## Statistical analysis

Graphs and statistics were done with SigmaStat12.3 (Systat Software GmbH, Frankfurt am Main, Germany). We aimed at minimum five individuals per group in order to run non-parametric Mann-Whitney rank sum tests where applicable. The group of non-primates was compared to macaque, human was not included in the tests. Source data for the graphs were included as excel files.

## Acknowledgements

PW and GM dedicate the paper to our friend and mentor Prof. Dr. Klaus Albus, who graciously declined to join as a coauthor although the developing cat material we investigated had been prepared in his lab at the Max-Planck-Institut für Biophysikalische Chemie, Göttingen, Germany. We thank Prof. Barry B Lee, at that time at the Max-Planck-Institut für Biophysikalische Chemie, Göttingen, Germany, for sharing monkey brain material. We thank Prof. Dr. Klaus-Peter Hoffmann, Ruhr-Universität, Bochum, Germany, who led the studies delivering the biocytin material of rat, cat, ferret and macaque. We thank Dr. Astrid Rollenhagen, JARA-Institute Brain Structure Function Relationship, Jülich, for advice with the human patient material.

## Additional information

### Funding

| Funder | Grant reference number | Author |
| --- | --- | --- |
| Deutsche Forschungsgemeinschaft | WA 541/13-1 | Petra Wahle |
| Deutsche Forschungsgemeinschaft | WA 541/15-1 | Petra Wahle |
| Deutsche Forschungsgemeinschaft | EN 1240/2-1 | Maren Engelhardt |
| Deutsche Forschungsgemeinschaft | Ho-450/25-1 | Claudia Distler |
| Deutsche Forschungsgemeinschaft | SFB 509/A11 | Claudia Distler |

The funders had no role in study design, data collection and interpretation, or the decision to submit the work for publication.

### Author contributions

Petra Wahle, Conceptualization, Formal analysis, Funding acquisition, Investigation, Methodology, Project administration, Resources, Supervision, Visualization, Writing - original draft, Writing - review

and editing; Eric Sobierajski, Ina Gasterstädt, Formal analysis, Investigation, Visualization; Nadja Lehmann, Susanna Weber, Investigation; Joachim HR Lübke, Resources; Maren Engelhardt, Formal analysis, Funding acquisition, Investigation, Resources, Writing - original draft, Writing - review and editing; Claudia Distler, Funding acquisition, Investigation, Resources, Writing - original draft, Writing - review and editing; Gundela Meyer, Formal analysis, Investigation, Resources, Writing - original draft, Writing - review and editing

**Author ORCIDs**
Petra Wahle http://orcid.org/0000-0002-8710-0375
Nadja Lehmann http://orcid.org/0000-0003-4801-3057
Joachim HR Lübke http://orcid.org/0000-0002-4086-3199

**Ethics**
EthicsThe data presented in this paper were collected via tissue sharing and from material that had originally been processed for projects not related to the present topic, i.e. no animals were sacrificed specifically for the present study.

**Decision letter and Author response**
Decision letter https://doi.org/10.7554/eLife.76101.sa1
Author response https://doi.org/10.7554/eLife.76101.sa2

---

## Additional files

**Supplementary files**
• Transparent reporting form

**Data availability**
All data generated or analysed during this study are included in the manuscript and supporting file; source data files have been provided for Figures 3 , 4, 5, 6, and 7.

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
