## [Editor Report]

Wahle and colleagues show that excitatory cortical neurons differ in the fundamental structural arrangement of dendrites, soma and axons across a range of mammalian species. Axons can originate directly from pyramidal cell dendrites in species as diverse as rodents, ferret, cats, pigs and primates. However, cross-species comparisons also indicate that non-primate brains have a higher proportion of axon-carrying-dendrites (AcD) than did brains of primates. This paper is of potential interest to a broad range of neuroscientists working on brain structure and function as well as computational models thereof.

---

## [Decision Letter]

**Decision letter after peer review:**

Thank you for submitting your article "Neocortical pyramidal neurons with axons emerging from dendrites are frequent in non-primates, but rare in monkey and human" for consideration by *eLife*. Your article has been reviewed by 3 peer reviewers, including Kristine Krug as Reviewing Editor and Reviewer #1, and the evaluation has been overseen by John Huguenard as the Senior Editor. The following individuals involved in review of your submission have agreed to reveal their identity: Kathleen Rockland (Reviewer #2); Jonathan B Levitt (Reviewer #3).

Essential revisions:

Your paper surveys an impressive range of species, in which you clearly show the presence of AcD. But all reviewers had some concerns about the comparability of the data from the different species, especially since to some extent (i) different neocortical areas were surveyed in different species and (ii) different histological methods were used in different species. In order to control for and frame these differences, we ask you for the following essential revisions:

1) At the beginning of the result, please, address explicitly the potential limits to the comparability of the data. Also, as far as possible, provide assessments to the following questions:

(a) to what extent is the proportion of neurons with AcD comparable within a given species across the different areas included, using any one of the anatomical methods employed. If there are differences, please give the range.

(b) to what extent is the proportion of neurons with AcD comparable within a given species and area using diverse methods (to provide evidence that proportion of AcD observed does not depend on the method used to reveal it)

2) Please clarify classification issues, i.e. whether shared-root always counted as AcD, and show whether this affected the data (perhaps in Figure 3: shaded bars of (AcD + shared roots) next to each bas of the result with just AcD)?

3) In the Methods, please provide more detail about sampling strategy (no. of sections, sampling windows, cells identified, layers covered).

4) In the Discussion, examine critically the limitations of your methods, especially to what extent area, tracer and sampling strategy mattered.

5) In the Abstract/Title, consider some qualification of the main finding of the results with regard the different areas investigated and range of methods used. The reviewers felt, as the manuscript stood, the clear-cut result was the wide-spread nature of AcD among pyramidal neurons, with the implication of lower proportions of AcD in primates being exciting in functional and evolutionary terms but which would benefit from deeper analysis and/or more data (see Point 1).

*Reviewer #1 (Recommendations for the authors):*

Wahle and colleagues surveyed the occurrence of the non-canonical of "axon carrying dendrites" of neocortical pyramidal cells across different mammalian species. Their results indicate that there are differences between rodents, pigs, cats and ferrets on the one hand and primates on the other. Pyramidal cells of primates (humans and monkeys) have overall a lower proportion of axon carrying dendrites in the gray matter, which seem to be more concentrated in infragranular layers and in white matter.

The major strength of the manuscript lie in the large scope of the systematic survey of pyramidal cells with axon carrying dendrites across a range of different mammalian species. The authors also include developmental data and results comparing different cortical layers as well as the results from white matter and gray matter. With the necessary diversity of histological source material necessary for this study, the authors should assess and discuss potential other sources of differences between the species and their strategy to overcome these more explicitly.

The presented data and scope of the study are impressive and shed light on the potential evolutionary differences in the cortical circuitry of primates in contrast to other mammals.

1) Lines 363-371: Classification of AcD and "shared roots". Please provide a clearer definition about the difference of these and AcD cells, perhaps a minimum requirement of dendritic length before the axon junction.

As other papers have included these in AcD cells, the discussion should include a short discussion of whether (or not) this could have affected differences with previous results or between species.

2) It would be good to see examples of interstitial AcD cells, perhaps in the supplement.

3) Discussion: This is quite brief and compacted.

a) I would suggest to pack and expand the discussion on development and function (p. 6 lines 204-2017) into to separate paragraphs.

b) Please put the proportions of cells found more explicitly into context with those reported in other papers, e.g. hippocampus in rodents, tufted pyramidal cells in layer 5 in rodents, and what we can learn from this for circuit function.

c) Please also discuss potential difference in classification (see also point 1) and sampling strategy (see also point 4) and their impact.

4) Methods lines 352ff: Please report sampling strategy (number of sections; size of field; how chosen).

5) p.3. Figure 1A-D: Figure legend and results text: the letter labelling got mangled in both.

6) Figure 3B: it would be easier for the reader if the identity line was shown.

7) Figure 4: y-axis has German-style commas rather than English-style points for the decimals. Please correct.

8) stat: lines 127-128 and line180-181. The latter references the statistics reported in the former. However, as written the precise comparisons made in lines 127-128 are not clear. Please rewrite.

9) There are a couple of places where the wording could be more precise and meaning should be checked:

– "non-primates" in abstract and else-where: I would suggest using more detailed descriptors, e.g. rodents, pigs, cats and ferrets or be clearer upfront that, the paper is concerned with mammals.

– line 192: "even more striking" – I do not to understand out of the context why this is the more striking result – please explain?

– line 201 " A clear majority …" suggests to me about 60-70%. I think with over 90%, it would be clearer to use "the vast majority" or " over 90%".

10) p 11. "Animal" section. Could the authors please summarise here all the species and numbers of animals for each species used, ages, and whether all experiments the tissue derived from conformed with the relevant welfare legislation.

*Reviewer #2 (Recommendations for the authors):*

This is an interesting and scientifically rigorous report documenting atypical, dendritic locations for the emerging axon of pyramidal neurons. This is not an entirely new observation (the authors cite relevant publications, including Kole and Brette, 2018 and Mendizabal-Zubiaga et al., 2007), but still important, as a relatively overlooked fact with functional implications. A main feature of the present report is an exceptionally thorough cross-species survey, from which the authors conclude that, as compared with non-primates, the macaque and human brains have a lower proportion of neocortical pyramidal neurons with axon carrying dendrites. The results might be further supported by additional experiments, especially ultrastructural data, or by including more extensive developmental data. There is a section on Development, but there is hardly any Discussion. However, these matters are raised and adequately treated by reference to the existing literature.

Improvements

1. The results on interstitial neurons should be supported by histology figures. Also, the authors could give more detail in the text; for example, is there a preferred spatial orientation relative to the overlying layer 6? Is this superficial white matter? Any preference for sulcal or gyral location?

2. The authors remark that AcD are "common in cortical inhibitory interneurons" (line 59), and quote several of their own previous papers. I suggest that this point could be elaborated in the Discussion, with commentary on why there are differences between excitatory pyramidal neurons and inhibitory interneurons. It would also be of interest if the authors have appropriate material to screen for AcD in calbindin-positive pyramidal neurons?

3. I think the textual presentation can be improved:

Abstract: line 43 talks about "much higher percentage" of AcD pyramidal cells in WM of postnatal cat and aged human. (1) Are there really "pyramidal neurons" in the WM (or: "pyramidal-like")? (2) line 48 has "mainly in deeper layers and white matter" of primates. The wording seems inconsistent with line 43, which does not include macaque….?

Line 44: The sentence on hippocampus seems out of place. This could better be developed in the Discussion.

Introduction. I think it would be more effective to highlight right off previous reports; i.e., as currently, Kole and Brette (references, but somewhat in the shadow of Cajal). Also, a better summary about previous reports of AcD in other populations, as inverted pyramidal neurons. Would the authors like to comment also on Von Economo cells (as, Banovac… Petanjek, 2021), or would these be considered interneurons??

Discussion. This section in particular seems overly succint and could be better organized. Lines 201-206 could be expanded, and seem to constitute a separate paragraph from neurons in the white matter. And a short paragraph might be added about hippocampal results. It might be useful to have again a short summary overview of the various populations having AcD (interneurons, inverted pyramids). Are there also unusual neurons with multi-axons?

*Reviewer #3 (Recommendations for the authors):*

The authors used neuroanatomical techniques to study neocortical pyramidal neurons from several different mammalian species. Their message is that primate neocortex differs from that of other mammals in having substantially fewer cells with axons emanating from dendrites, rather than the canonical route from the soma. The authors employed a range of standard methods, ranging from tracer injection to Golgi impregnation to immunocytochemistry. The feature the authors report is undeniable; there clearly are axons that emanate from dendrites of neocortical pyramidal neurons. Prior studies have reported that these axons are more excitable, thus leading to the intriguing possibility of a fundamental architectural (and thus presumably functional) feature in how primate neocortex operates.

This is a provocative narrative, that leads to a number of interesting questions. However, I have reservations that the authors must address before I believe the claim that primates are really fundamentally different from other mammals in this respect.

A strength but also a central limitation of this study is that different species were compared using different methods, and different areas were studied in different species. The authors make the implicit assumption that the prominence of this feature does not differ among cortical areas. However, it is entirely plausible that the proportion of neurons with axon-carrying dendrites does differ among cortical areas. The authors also group neurons into 2 large populations: infra- and supragranular. But again, layers 2 and 3 differ from one another (as do layers 5 and 6) in the specific populations of pyramidal cells they contain (morphological and neurochemical types, inputs and outputs, etc.). Certainly many studies do group neurons into these broad populations, but for this kind of comparison relevant differences or similarities could have been lost. Comparisons among species ideally would have all been in the same layer and area.

Another limitation is that the same method was not employed in different species. The reader needs to know that different methods reveal the same proportion of axon-carrying dendrites in a given area of a certain species. This should have been stated more clearly and earlier in the text; it took examination of the data tables to see this. The tables show that measurements were made in several different cortical areas. Can the authors provide any evidence that the proportion of neurons with axon-carrying dendrites does not differ in any one species among cortical areas? Figure 3 description and/or legend needs to state clearly that different species' neocortex was studied in different areas (and if all Figure 3 samples shown are from same layers). Supplementary Excel file suggests that for humans Golgi-Kopsch reveals fewer infragranular AcD-cells than Golgi-Cox (4.43 vs 1.39), while for adult macaques Golgi-Kopsch revealed fewer than biocytin injection or SMI-32/BetaIV-spectrin immunofluorescence (13.34 vs 7.98 vs 6.29). Since the human data relies on Golgi methods, the authors must reassure the readers that the comparison of species is validated by direct comparison of different methods.

The message that primates have fewer cells with axon-carrying dendrites than other mammals might therefore certainly be interesting but far less compelling. The message might be that primate neocortex is not qualitatively different from that of other species; instead they simply have somewhat fewer AcD-bearing neurons than other mammalian species. But even that more modest conclusion is suggested but not fully proven by the data here.

I was puzzled by Figure 4 not including primate tissue. If the message is that spine density does not differ in dendrites with and without axons, surely it would be important to include primate tissue in this comparison; the comparison between primates and on-primates is after all the core message of this study. I also do not think the values for each species for non-AcD and shared root should be connected by a line; I suggest instead there should simply be a scatter of values for each group with a large symbol indicating mean or median value of each group. This would facilitate comparison.

This study would be more convincing if the authors could provide more data showing both:

i) Proportion of neurons with AcD is the same in a given species in multiple areas, using any one of the anatomical methods employed (to prove all cat or ferret or monkey areas are equivalent).

ii) Proportion of neurons with AcD is the same in a given species and area using multiple methods (to prove that proportion observed does not depend on method used to reveal it).

---

## [Author Response]

Essential revisions:Your paper surveys an impressive range of species, in which you clearly show the presence of AcD. But all reviewers had some concerns about the comparability of the data from the different species, especially since to some extent (i) different neocortical areas were surveyed in different species and (ii) different histological methods were used in different species. In order to control for and frame these differences, we ask you for the following essential revisions:1) At the beginning of the result, please, address explicitly the potential limits to the comparability of the data. Also, as far as possible, provide assessments to the following questions:

Yes, our aim has been to provide a survey. We rather saw a strength in the fact that we were not looking just into one cortical area, and not only with one method. However, we agree that this might have appeared a bit “unsystematic” to the readers. We understand that referees immediately ask for a more systematic analysis. The reviewers may have noted that we analyzed archived material, not killing a single animal for the present study. This implies that we are bound to what we have, in particular, for the human material.

Throughout the paper we have now critically compared the methods to point out potential limitations. We enlarged the methods section to be clear on how we assessed the material. We hope that the referees will find this convincing. We are of course willing to enlarge on additional points should those occur.

(a) to what extent is the proportion of neurons with AcD comparable within a given species across the different areas included, using any one of the anatomical methods employed. If there are differences, please give the range.(b) to what extent is the proportion of neurons with AcD comparable within a given species and area using diverse methods (to provide evidence that proportion of AcD observed does not depend on the method used to reveal it)

Within-species comparison: For macaque we have massively expanded out analysis choosing the most reliable tools, SMI-32/βIV-spectrin labeling of the AIS plus confocal analysis with cryosections of three individuals in which the immuno worked nicely. We now show with a direct comparison in the new Figure 4A that visual, auditory, somatosensory, cingulate, intraparietal and parietal/premotor cortex display 3-7% AcD neurons when sampled through all layers. The limbic (iso-)cortex did not differ recognizably from the sensory (iso-)cortices. The average ranged from 3.78 ± 0.85% in V1 to 5.63 ± 1.51% in cingulate cortex. At this moment we have to accept a variability of areas of a factor of 2, which is about the interindividual variability in macaque and also in the non-primates. We believe this is a rather small variability. Proportions published with intracellular labeling methods recently in mouse CA1 neurons vary from 20% (Benavides-Piccione et al., 2020) to >50% (Thome et al., 2014).

Within-species comparison: We added another 5 cats with striate and extrastriate biocytin injections which allowed a comparison between striate (area 17) and extrastriate cortex (areae 18+19). There is no systematic difference in the proportion of AcD neurons.

Across-species comparison: From the data presented in the Tables 1-4 (we had to split one table to incorporate more human data) readers can compare across areas and between species. The ferret data are “visual” cortex and absolutely comparable to the cat; just the injection site was in extrastriate cortex (area 18). We corrected this to “visual cortex” in Table 2. Also in Figure 4 we directly compared visual cortex of cat (n=7) and ferret (n=4). The medians were nearly identical. From all this it is unlikely that the proportion of AcD (defined by strict criteria!) observed depends on the method used to reveal it when comparing the intracellular staining methods, biocytin and fluorescence.

Across-methods comparison: The referees were right at this point. Things might differ when comparing across methods, which unfortunately is rarely found in the literature. In Figure 3B we assessed the laminar proportions. Non-primates and macaque differed significantly. We no longer reported a statistical comparison of macaque and human in Figure 3B. We indeed got the impression that the Golgi method slightly underrepresented the AcD neurons and yielded a higher proportion of shared root cells. The difficulty in deciding WHERE exactly the axon originates becomes clear with the new Figure 2—figure supplement and Figure 2—figure supplement video 2. Overall the proportion of shared root cells varied between areas and individuals by a factor of 10 whereas the clear-cut AcD cells varied only by a factor of about 2 (e.g. in macaque from 3% to ~7%)!

One referee asked about Golgi-Cox versus Golgi-Kopsch. Methods have traditions in the labs worldwide, and so, methods have often not been directly compared within one lab. As one example we suggest Banovac et al., (2019) who compared two Golgi methods finding that both delivered the same results.

Actually, an overestimation of shared root cells has also emerged with the biocytin data, but here, it was only for detection of shared root cells, and did not affect the detection of clear-cut AcD cells. This has now been addressed in the new Figure 5C.

Within-individual comparison: For macaque #2 (his name was Platon) we could obtain in total six values with immunofluorescence from the various areas, plus five values from the biocytin material (see source data). Again, detection of AcD cells was equally good with both methods, but shared root cells were a bit overrepresented. This has now been addressed in the new Figure 5D.

We now write in Results:

“Next, for rat, ferret, macaque and human, we compared the percentages of AcD to the sum of AcD plus shared root (Figure 5B). If the shared root cells were considered as AcD cells, the proportions of AcD cells increase to some extent in all species analyzed. The interindividual variability of the shared root cells was at a factor of 10 (range of 0.46% to 5.5% in macaque), and statistics argued against any biologically significant difference between species.

Unexpectedly, a subtle difference was observed independently by two observers who analyzed the Golgi material (PW at Ruhr University Bochum, GM at University La Laguna). 13 of 13 cases (2 macaque, 11 human individuals) had percentages of shared root cells higher than percentages of AcD cells, whereas in 22 of 25 individuals and/or cortical areas stained for biocytin and immunofluorescence the percentages of shared root cells were lower than the percentages of AcD cells (Figure 5B). Thus, it might be that the proportion of AcD neurons was slightly underrepresented in our Golgi material. Yet, also the biocytin material had larger proportions of shared root cells (Figure 5B). We therefore compared immunofluorescence and biocytin of the macaque material (Figure 5C). Indeed, the biocytin material delivered significantly more shared root cells, whereas the unequivocal AcD cells were equally well recognized. Of one macaque, Individual 2, we could assess a number of regions (see source data) allowing to compare the two methods within one individual (Figure 5D). Again, the proportion of shared root cells in the biocytin material was higher than with immunofluorescence whereas unequivocal AcD again were detected equally well with the two methods.”

We now discuss the issue as follows:

“We did not run statistical comparisons with our human data. After analyzing more and more individuals and/or brain areas it became evident that the Golgi methods yielded a lower proportion of AcD neurons and the higher proportion of shared root cells. In line with this, also the biocytin material yieded higher proportions of shared root cells. A parsimonious explanation may be as follows. The Golgi reaction product is a chromate precipitate deposited at the plasma membrane. The pitch-black reaction product, the thickness of the tissue sections, on top of the complexity of basal dendrites in primate (Hendry and Jones, 1983) and even more so in human (Mohan et al., 2015; review by Goriounova and Mansvelder, 2019) can make it difficult to determine if an axon emerges from a soma, or from a shared root, or already from a very proximal dendritic trunk. The same accounts for the dark biocytin reaction product although the section thickness here was thinner. An additional argument comes from the axon itself. Axons originating from dendrites are thinner and have less prominent hillocks (Peters et al., 1968; Mendizabal-Zubiaga et al., 2007; Benavides-Picchione et al., 2020). With dark reaction products it was difficult to precisely determine where exactly a thin process lacking a clear hillock arises from a large dendritic root. This way we counted somewhat higher percentages of shared root and somewhat lower percentages of AcD in the Golgi Cox and Golgi-Kopsch material. By contrast, the intracellular staining of much thinner sections such as the 20-50 µm thick sections of the biocytin and immunofluorescence material allowed to visualize structures at better optical resolution. In particular, the confocal analysis allowed to walk micrometer-by-micrometer through the optical stack to decide “pro AcD” or “pro shared root” for each case in question arguing that the optical resolution was the crucial parameter. Nevertheless, biocytin staining was equal to the immunofluorescence in detecting clear-cut AcD, but was inferior to immunofluorescence and confocal analysis when it comes to decide on shared root. It should be noted that the frequently used SMI-32 staining method may also have a certain bias in that it stains preferentially type 1 pyramidal neurons (Molnar et al., 2006). Future studies are needed before a final conclusion on the areal and laminar proportion of human pyramidal AcD neurons can be made, and for a species comparison intracellular staining methods should be applied as recently done for CA1 pyramidal cells (Benavides-Picchione et al., 2020).”

2) Please clarify classification issues, i.e. whether shared-root always counted as AcD, and show whether this affected the data (perhaps in Figure 3: shaded bars of (AcD + shared roots) next to each bas of the result with just AcD)?

Shared root cells have consequently be included in the group of somatic cells. We accepted as AcD cells only clear-cut examples! This is stated several times in Methods and in Results, and in Discussion. We now deliver in Figure 5 a whole new analysis, triggered by the impression that the Golgi method led to an overestimation of the shared root cells and underestimation of AcD cells in macaque as well as in human.

We provide two additional color Figures (supplements to Figure 2) with cells from these animals and cells from the human material to better clarify what we consider a shared root and an AcD versus a somatic axon. For mice, appropriate pictures are in the literature. In other aspects, our answer to this point overlaps with the answer to Point 1. What can be easily seen is how difficult it is with the Golgi material to determine if a thin axon is from a dendrite of still from the soma. Believe us, we photographed cases where the picture was fairly clear. There are cases where regular photography is no longer able to resolve issues, but focus drive and the trained observer’s retina and brain are able to make decisions.

It is a matter on how strictly the observer team implements the criteria. We submit to you an old school blackboard sketch done to train new students in the lab. S, somatic axon cells (Author response image 1). The SR, shared root is a gradual transit of the axon hillock towards a dendrite. As long as a vector through the hillock points into the soma we scored the neurons as shared root, and added these cases to the group of somatic axon cells.

**Author response image 1. sa2fig1:** 

At this point one may wonder if computerized approaches could do any better. We believe: NO! Automated systems are no better (even worse in my experience) than trained neuroanatomists, for instance in segmenting a neuron and delineating soma, dendrite and axon (see for instance Luengo-Sanchez S et al., Front Neuroanat 2015, doi: 10.3389/fnana.2015.00137)After all, we felt we adopted the right strategy in counting ONLY the clear-cut AcD cells and not a mix of clear-cut AcD cells plus shared root cells. This has been done in many papers which seemingly were more relaxed on definitions. Had we done so, it would have increased the interindividual variability, the laminar variability, and the interareal variability substantially. We have to await electrophysiological data in order to see whether there are functional reasons for splitting or for lumping together true AcD and shared root cells.

3) In the Methods, please provide more detail about sampling strategy (no. of sections, sampling windows, cells identified, layers covered).

We have done so in the method section, as well as giving all details on sampling in the source data. Methods have been extended. For instance, we describe that we assessed a substantial part of the animal material actually 3 (!) times, first using a quite naïve strategy for total proportion (perpendicular tracks pia to white matter). Later, when looking at the macaque we learned that laminar percentages can differ. Consequently, we counted again in surface-parallel tracks. Finally, on your request, we assessed the shared root cells in the animal material because those were scored as “somatic” in the first rounds of counting. Please see the source data for every Figure: it now has the number of sections, the spacing, the thickness, the number of confocal stacks and the total number of neurons assessed with each round of counting in every area and individual. One can regard this almost as in “internal control”, as we again and again ended with very similar percentages for every individual, always landing within the range of that species. The repeat counts explain why the percentages plotted in the Figures to the various aspects are not exactly identical.

4) In the Discussion, examine critically the limitations of your methods, especially to what extent area, tracer and sampling strategy mattered.

The answer overlaps with that of Point 1. Please let us know if you feel that this is not sufficient or if we had overlooked other limitations.

5) In the Abstract/Title, consider some qualification of the main finding of the results with regard the different areas investigated and range of methods used. The reviewers felt, as the manuscript stood, the clear-cut result was the wide-spread nature of AcD among pyramidal neurons, with the implication of lower proportions of AcD in primates being exciting in functional and evolutionary terms but which would benefit from deeper analysis and/or more data (see Point 1).

We revised the abstract accordingly.

Since one of the reviewers mentioned interneurons, we are happy to offer a substantial data set on non-pyramidal neurons in human (Golgi, Parvalbumin) and in cat (neuropeptide immunolabeling) in the new Figure 7A, and 7B. It shows that the proportion of AcD is a specific feature of the cell types. This confirms previous interpretations of material in rodent (Höfflin et al., 2017).

The Golgi-impregnated subset of interneurons in upper layers of human cortex includes bitufted neurons with arcade axons, Martinotti neurons with ascending axons, and axons resembling local small basket cells. Due to the age of the individuals and the full myelination, only the initial portions of the axons were impregnated. These types are equivalent to Somatostatin-ir neurons of the non-fast-spiking class in other mammals. To our surprise the interneurons in human behave as they do in non-human mammals: many have axons from dendrites whereas human upper layer pyramidal cells rarely have AcD. Further, we provide an assessment of AcD for Parvalbumin-positive neurons in human. They have rather low proportions of AcDs. So, the feature segregates with neuron type. To this end it is suggestive to assume that the AcD phenotype is a regulated feature.

We provide a graph with AcD quantification of interneuron types of developing cat visual cortex (striate and extrastriate) including NPY-ir axonal loop cells of the subplate/white matter (the WM neurons can’t be allocated to any particular area, the neurons project all over), of NPY-ir Martinotti and small basket cells of gray matter layer VI, and of Somatostatin-positive Martinotti and bitufted cells of gray matters VI and V and upper layers from the third week onwards.

Our study cannot answer the question of HOW the axon ends up on a dendrite. The in vitro literature on dissociated cells is full of funny pictures of “multiple-axon-cells”. Neuronal polarity might entirely be messed-up with culturing; so, we do not learn much from this. For the in vivo situation, one could imagine that the axon is “passively pulled away” from the soma in the process of basal dendritic growth. The shared root configuration may be seen as in-between state. Yet, this can’t explain why it seemingly happens significantly less often in primate isocortex, but frequently in hippocampus! As stated above, the AcD phenotype segregates with neuron type. Should we think of specific molecular mechanisms which keep the axon on the soma? Equally well, however, the AcD neurons in primates might became eliminated during early developmental stages by some form of activity-dependent cell death. The questions for future studies are first, is it a regulated process? And if yes, what does a network and eventually the behaving animal gain from harboring (or not harboring!) a certain proportion of AcD pyramidal cells? Of course, this is just some thoughts, and we have not included such an outlook in the manuscript.

Reviewer #1 (Recommendations for the authors):1) lines 363-371: Classification of AcD and "shared roots". Please provide a clearer definition about the difference of these and AcD cells, perhaps a minimum requirement of dendritic length before the axon junction.As other papers have included these in AcD cells, the discussion should include a short discussion of whether (or not) this could have affected differences with previous results or between species.

See our answer to Point 3 and the picture provided. We tried to clarify definitions much better. At this moment there is no agreement in the community towards “what is an AcD cell” and “what is a shared root cell”, published data between labs cannot be exactly compared because definitions are not always identical.

2) It would be good to see examples of interstitial AcD cells, perhaps in the supplement.

It is published. See Meyer G, Wahle P, Castaneyra-Perdomo A, Ferres-Torres R (1992) Morphology of neurons in the white matter of the adult human neocortex *Experimental Brain Research* 88:204-212. https://doi.org/10.1007/BF02259143

The funny issue here is that we over the years have so often written (as many others) “…axons emerge from the soma or one of the dendrites….” Just that in the old times nobody has thought on the functional implications.

3) Discussion: This is quite brief and compacted.a) I would suggest to pack and expand the discussion on development and function (p. 6 lines 204-2017) into to separate paragraphs.b) Please put the proportions of cells found more explicitly into context with those reported in other papers, e.g. hippocampus in rodents, tufted pyramidal cells in layer 5 in rodents, and what we can learn from this for circuit function.c) Please also discuss potential difference in classification (see also point 1) and sampling strategy (see also point 4) and their impact.

We have done so. With so many new data, analyzes and Figures, the Results and Discussion part has changed a lot. We also included numerical comparisons to published data on AcD cells, which actually are quite discrepant between labs (see Benavides-Piccione et al., 2020 versus Thome et al., 2014).

We refrained from discussing functional implications. Our study is a morphological assessment.

4) Methods lines 352ff: Please report sampling strategy (number of sections; size of field; how chosen)

All this is now meticulously listed in the source data, all include lists of the histological material that we assessed.

5) p.3. Figure 1A-D: Figure legend and results text: the letter labelling got mangled in both.

This has been repaired, thanks for notifying.

6) Figure 3B: it would be easier for the reader if the identity line was shown.

We could have done this, but we felt that one should not overdecorate a Figure with regressions. From a mathematical perspective one would wish for more individuals here to place identity lines for the each taxon. It might need substantial amounts of animals, though. If the Editors wish, we can so for the macaque cluster and the non-primate cluster. We leave it to you.

7) Figure 4: y-axis has German-style commas rather than English-style points for the decimals. Please correct.

This has been repaired, thanks for notifying.

8) stat: lines 127-128 and line180-181. The latter references the statistics reported in the former. However, as written the precise comparisons made in lines 127-128 are not clear. Please rewrite.

This has been done. Given the possibility (with more human data) of an underrepresentation of AcD cells detected with the Golgi method, we decided to no longer include statistics on human versus macaque since it is at this moment not clear it the human values are really lower than those of macaque or overlapping with the lower macaque values.

9) There are a couple of places where the wording could be more precise and meaning should be checked:– "non-primates" in abstract and else-where: I would suggest using more detailed descriptors, e.g. rodents, pigs, cats and ferrets or be clearer upfront that, the paper is concerned with mammals.

It has been listed already in the abstract : “Here, we report on the diversity of axon origins in neocortical pyramidal cells. We found that in non-primate mammals (we assessed mouse, rat, cat, ferret, pig),…”

– line 192: "even more striking" – I do not to understand out of the context why this is the more striking result – please explain?– line 201 " A clear majority …" suggests to me about 60-70%. I think with over 90%, it would be clearer to use "the vast majority" or " over 90%".

We have been more careful with “extremes”. Thanks.

10) p 11. "Animal" section. Could the authors please summarise here all the species and numbers of animals for each species used, ages, and whether all experiments the tissue derived from conformed with the relevant welfare legislation.

It is all in source data, and the histological material is from published papers where the legal issues accounting at those times are mentioned.

Reviewer #2 (Recommendations for the authors):This is an interesting and scientifically rigorous report documenting atypical, dendritic locations for the emerging axon of pyramidal neurons. This is not an entirely new observation (the authors cite relevant publications, including Kole and Brette, 2018 and Mendizabal-Zubiaga et al., 2007), but still important, as a relatively overlooked fact with functional implications. A main feature of the present report is an exceptionally thorough cross-species survey, from which the authors conclude that, as compared with non-primates, the macaque and human brains have a lower proportion of neocortical pyramidal neurons with axon carrying dendrites. The results might be further supported by additional experiments, especially ultrastructural data, or by including more extensive developmental data. There is a section on Development, but there is hardly any Discussion. However, these matters are raised and adequately treated by reference to the existing literature.

We cannot do EM with frozen material or DEPEX-cleared sections. The developmental aspects have been more extensively discussed now, but we refrained from speculating too much, since we do not have physiological data.

Improvements1. The results on interstitial neurons should be supported by histology figures. Also, the authors could give more detail in the text; for example, is there a preferred spatial orientation relative to the overlying layer 6? Is this superficial white matter? Any preference for sulcal or gyral location?

Please see above, where we copied on the Figures by Meyer et al., 1992 into the rebuttal. Human white matter pyramidal cells are in the gyral white matter oriented vertically but also occur millimeters away from the gray matter, buried deep in the fiber tracts. And they happily survive because we saw them in >80-year old individuals.

2. The authors remark that AcD are "common in cortical inhibitory interneurons" (line 59), and quote several of their own previous papers. I suggest that this point could be elaborated in the Discussion, with commentary on why there are differences between excitatory pyramidal neurons and inhibitory interneurons. It would also be of interest if the authors have appropriate material to screen for AcD in calbindin-positive pyramidal neurons?

We refer this reviewer to the new Figure 7A,B, and the new chapter on interneurons. Quite a lot has been published to that point, but in past has been anecdotal because years ago nobody thought about the functional implications. We (G.M.) actually tried to screen calbindin-positive neurons in human, but the labeling was so dense – no safe assessment possible. It would require fresh material with double-labeling. At this moment we have to leave a thorough investigation of the human material with additional methods to the future.

3. I think the textual presentation can be improved:Abstract: line 43 talks about "much higher percentage" of AcD pyramidal cells in WM of postnatal cat and aged human. (1) Are there really "pyramidal neurons" in the WM (or: "pyramidal-like")? (2) line 48 has "mainly in deeper layers and white matter" of primates. The wording seems inconsistent with line 43, which does not include macaque….?Line 44: The sentence on hippocampus seems out of place. This could better be developed in the Discussion.

See above, Meyer et al., 1992. By classical criteria (polarity, a major dendrite, spines) these cells are comparable to pyramidal cells of layer VI. The paper by Friauf and Shatz (quoted) has looked at these cells with intracellular staining and recordings – they are pyramidal by all criteria.

Introduction. I think it would be more effective to highlight right off previous reports; i.e., as currently, Kole and Brette (references, but somewhat in the shadow of Cajal). Also, a better summary about previous reports of AcD in other populations, as inverted pyramidal neurons. Would the authors like to comment also on Von Economo cells (as, Banovac… Petanjek, 2021), or would these be considered interneurons??

We have quoted the Banovac et al., review, in order to not include too much speculations in our manuscript. We did in fact see similar fusiform neurons also in human temporal cortex, but the sample size is too small to make an extra story on VENs. We encountered one cell resembling a VEN in cingulate cortex, and this cell can now be viewed in video 3.

Discussion. This section in particular seems overly succint and could be better organized. Lines 201-206 could be expanded, and seem to constitute a separate paragraph from neurons in the white matter. And a short paragraph might be added about hippocampal results. It might be useful to have again a short summary overview of the various populations having AcD (interneurons, inverted pyramids). Are there also unusual neurons with multi-axons?

The first submission has been for a Short Report and given the word limits, we had to stay succinct and short. With the new data included in the revision, also the Discussion has been enlarged. Hippocampal findings have been quoted, for instance. And yes, there are interneurons with two axons, but we found only a handful. We scored them in the neuropeptide-stained material when both axons originated from the soma or both from dendrites. We ignored them, when the axons originated from soma and a dendrite…. to which group should they go?

Reviewer #3 (Recommendations for the authors):The authors used neuroanatomical techniques to study neocortical pyramidal neurons from several different mammalian species. Their message is that primate neocortex differs from that of other mammals in having substantially fewer cells with axons emanating from dendrites, rather than the canonical route from the soma. The authors employed a range of standard methods, ranging from tracer injection to Golgi impregnation to immunocytochemistry. The feature the authors report is undeniable; there clearly are axons that emanate from dendrites of neocortical pyramidal neurons. Prior studies have reported that these axons are more excitable, thus leading to the intriguing possibility of a fundamental architectural (and thus presumably functional) feature in how primate neocortex operates.This is a provocative narrative, that leads to a number of interesting questions. However, I have reservations that the authors must address before I believe the claim that primates are really fundamentally different from other mammals in this respect.A strength but also a central limitation of this study is that different species were compared using different methods, and different areas were studied in different species. The authors make the implicit assumption that the prominence of this feature does not differ among cortical areas. However, it is entirely plausible that the proportion of neurons with axon-carrying dendrites does differ among cortical areas.

We initially considered it a strength of the study – looking into many area with many methods in many species. However, it seemed a bit like cherry-picking, and we now enlarged the data sets for a more systematic analysis. Please note, we assessed archived material. We are bound to what we have available. We now delivered areal comparisons, and I am afraid, the answer is NO, no remarkable differences in the areas that we assessed in monkey and cat.

The authors also group neurons into 2 large populations: infra- and supragranular. But again, layers 2 and 3 differ from one another (as do layers 5 and 6) in the specific populations of pyramidal cells they contain (morphological and neurochemical types, inputs and outputs, etc.). Certainly many studies do group neurons into these broad populations, but for this kind of comparison relevant differences or similarities could have been lost. Comparisons among species ideally would have all been in the same layer and area.

As said, we are bound to what we have available. And this is more than what has ever been published on these question so far. The graph and the Tables to Figure 3B allow to compare species across the layers.

We are aware that pyramidal cells in the layers can differ. Looking into RNA seq papers, up to 19 types exist in mouse. How many could potentially then exist in human? There is no way of pulverizing our kind of analysis down to the level of 19 pyramidal cell types differing by some unexplained RNA signatures which so far exist only for mouse. The SMI-32 staining already “selects” for one subtype in that it stains preferentially so-called type 1 pyramidal cells (Molnar et al., 2006).

Another limitation is that the same method was not employed in different species. The reader needs to know that different methods reveal the same proportion of axon-carrying dendrites in a given area of a certain species. This should have been stated more clearly and earlier in the text; it took examination of the data tables to see this. The tables show that measurements were made in several different cortical areas. Can the authors provide any evidence that the proportion of neurons with axon-carrying dendrites does not differ in any one species among cortical areas?

We now provide areal comparisons for 5 fields in monkey (new Figure 4A) and visual fields in cat (new Figure 4B), both with the same methods. We can even provide a within-individual comparison of brain areas and of methods. Another three areal values for the infant macaque have been plotted in Figure 3B.

Figure 3 description and/or legend needs to state clearly that different species' neocortex was studied in different areas (and if all Figure 3 samples shown are from same layers).

Figure 3A is total cortex, Figure 3 B is by layers. Counting strategies are now described in detail in methods.

Supplementary Excel file suggests that for humans Golgi-Kopsch reveals fewer infragranular AcD-cells than Golgi-Cox (4.43 vs 1.39), while for adult macaques Golgi-Kopsch revealed fewer than biocytin injection or SMI-32/BetaIV-spectrin immunofluorescence (13.34 vs 7.98 vs 6.29). Since the human data relies on Golgi methods, the authors must reassure the readers that the comparison of species is validated by direct comparison of different methods.The message that primates have fewer cells with axon-carrying dendrites than other mammals might therefore certainly be interesting but far less compelling. The message might be that primate neocortex is not qualitatively different from that of other species; instead they simply have somewhat fewer AcD-bearing neurons than other mammalian species. But even that more modest conclusion is suggested but not fully proven by the data here.

The referee was right at this point. Please see the full answer to this question above (essential points for authors). Having doubled our data sets with more human data we now agree: the Golgi method underestimates the AcD neurons simply because of optical limitations. We now extensively discuss the issue and we no longer do statistical analysis on human. The issue needs further investigation with more methods.

I was puzzled by Figure 4 not including primate tissue. If the message is that spine density does not differ in dendrites with and without axons, surely it would be important to include primate tissue in this comparison; the comparison between primates and on-primates is after all the core message of this study. I also do not think the values for each species for non-AcD and shared root should be connected by a line; I suggest instead there should simply be a scatter of values for each group with a large symbol indicating mean or median value of each group. This would facilitate comparison.

First to the graph on spines, now Figure 6. You have to connect the individual neurons by line, otherwise the major point can no longer be seen: the dendrites differ in spine counts, sometimes the AcD is higher than the other basals of the very same neuron, in the next cell the AcD had a lower count. Statistics did not even suggest a trend. We agree that things may differ in immature neurons. Possibly, during early development the AcD gains advantages by means of its higher excitability.

Please read the methods part to this point, eligible neurons had to fulfill a number of criteria. We fully exploited the available material of rat and ferret; no more eligible neurons. We indeed tried the same in macaque. Section thickness 50 µm. We found exactly two neurons which fulfilled the criteria. We had no chance with this material given the enormous dimension of the pyramidal cell dendritic trees in monkey. They were simply cut. For this type of classical tracing studies, non-alternating section series were prepared and submitted to different types of staining. Section spacing was several hundred µm in each individual. No chance to “reconstruct” dendrites from adjacent sections, since there were no adjacent sections.

The core message of the study is still valid, also without the spine analysis in monkey.

This study would be more convincing if the authors could provide more data showing both:i) proportion of neurons with AcD is the same in a given species in multiple areas, using any one of the anatomical methods employed (to prove all cat or ferret or monkey areas are equivalent).

Please see the new Figure 4A, B.

ii) proportion of neurons with AcD is the same in a given species and area using multiple methods (to prove that proportion observed does not depend on method used to reveal it).

Please see the new Figure 4A, B. Please also see the within-individual comparison in Figure 5D.